# Topology-Aware Contrastive Learning: Regulating Representation Connectivity via Persistent Homology

**Jiaxin Sun** [1 2]   **Yuhua Qian** [1 2 3]   **Yang Wang** [1 2]

## Abstract

Standard contrastive learning minimizes geometric distance between positive pairs, implicitly assuming that strict compactness optimizes discrimination. However, this *topology-agnostic confusion* neglects intrinsic data structures and topological complexity, leading to class confusion—particularly when aggressive augmentations induce semantic drift. To address this, we propose *Topology-Aware Contrastive Learning*, a framework that shifts the objective from geometric singularity to topological connectivity. Leveraging *Persistent Homology*, we explicitly regulate the connectivity of the latent space, ensuring positive pairs maintain an $\alpha-\beta$ *connectivity* that balances intra-class cohesion with separability. Theoretically, we formally define the topology-agnostic confusion phenomenon, prove that excessive compactness strictly lower-bounds the probability of confusion and derive a generalization bound demonstrating that richer topological connectivity tightens downstream risk. Furthermore, we establish a measure-theoretic framework to mitigating the sensitivity of our method against varying augmentation strengths. Empirical results on benchmarks confirm that our approach enhances representation quality and reduces reliance on specific augmentation strategies compared to standard baselines.

## 1. Introduction

In recent years, the rapid advancement of self-supervised learning has reignited interest in contrastive learning (He

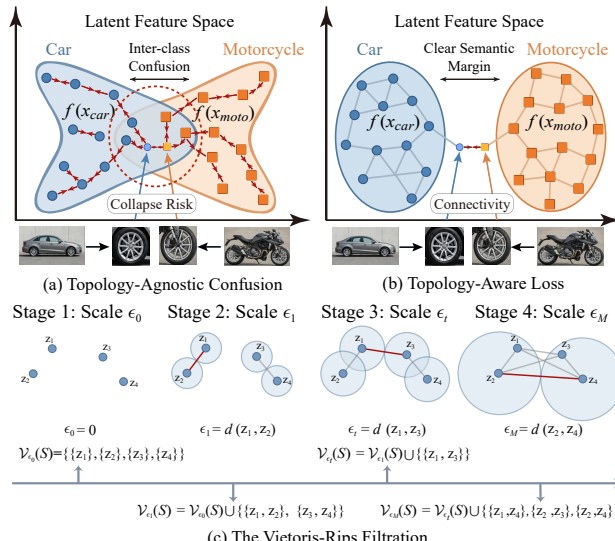

*Figure 1.* Schematic of latent topological structures. (a) Visualization of topology-agnostic confusion in standard contrastive learning, highlighting manifold entanglement and collapse risks. (b) The resulting structure from our topology-aware loss, featuring preserved intra-class connectivity and clear semantic margins. (c) Illustration of the Vietoris-Rips filtration process showing the evolution of connected components across increasing scales $\epsilon$.

et al., 2020; Oord et al., 2018; HaoChen et al., 2021; Caron et al., 2020; Chen & He, 2021; Grill et al., 2020; Zbontar et al., 2021). The standard framework, exemplified by objectives such as InfoNCE (Chen et al., 2020; He et al., 2020; Poole et al., 2019), operates on a core geometric principle: identifying invariances by minimizing the distance between augmented views of the same instance while pushing apart distinct instances. This approach incentivizes the encoder to map positive pairs to identical points within the embedding space (Saunshi et al., 2019; Wang & Isola, 2020).

However, strictly enforcing such geometric proximity neglects the inherent topological complexity of the data, leading to detrimental over-alignment and the distortion of the latent semantic structure (Wang & Liu, 2021; Huang et al., 2021). This issue becomes critical when strong data augmentations induce semantic drift (Zhang & Chen, 2025). Consider a motivating example: an image of a "car" and a

[1]Institute of Big Data Science and Industry, Shanxi University, Taiyuan, China [2]Key Laboratory of Evolutionary Science Intelligence of Shanxi Province, Shanxi University, Taiyuan, China [3]School of Artificial Intelligence, Shanxi University, Datong, China. Correspondence to: Yuhua Qian <jinchengqyh@126.com>.

*Proceedings of the 43[rd] International Conference on Machine Learning*, Seoul, South Korea. PMLR 306, 2026. Copyright 2026 by the author(s).

"motorcycle". Strong augmentations might crop both to focus solely on a "wheel". Since the contrastive loss forces the augmented view (wheel) to align strictly with its anchor, the encoder is incentivized to map distinct classes dangerously close, creating spurious bridges between semantic clusters. We refer to this phenomenon as "topology-agnostic confusion", where representations that should remain separable become inadvertently entangled (see Figure 1(a)).

To mitigate this, we propose a fundamental shift from strict geometric collapse to topological connectivity. We posit that for a robust representation, positive pairs do not need to be geometrically identical; rather, they effectively need to be topologically connected—falling within the same connected component of the latent structure. By maintaining a controllable topological distance, we preserve the distinct structures of different classes without aggressively collapsing the space (see Figure 1(b)).

To operationalize this, we introduce a Topology-Aware Contrastive Learning framework. Our key insight is to leverage Persistent Homology (Chen et al., 2019; Hofer et al., 2019) to explicitly regulate the connectivity of representations. We augment the contrastive objective with a topological term that ensures positive pairs are brought within a target distance to guarantee intra-class connectivity, while simultaneously preventing the formation of cross-class shortcuts. Through this lens, the aim is no longer to force invariance indiscriminately, but to sculpt a representation space whose topology reflects the latent semantic organization of the data.

Our theoretical analysis transcends empirical improvements, establishing a rigorous theoretical framework that links topological constraints to generalization capabilities. We derive excess risk bounds for both pre-training and downstream tasks, demonstrating that regulating connectivity directly tightens error bounds. Furthermore, we address the sensitivity of contrastive learning to data augmentation through unified theoretical and empirical validation.

In summary, our main contributions are as follows:

- We propose the first contrastive objective that integrates Persistent Homology to explicitly regulate latent connectivity. By controlling connected components, our method enforces intra-class linkage while precluding spurious cross-class bridges, effectively averting both feature collapse and inter-class confusion.

- We analyze the contrastive mechanism through the lens of connectivity, introducing $\alpha$–$\beta$ connectivity to formalize how positive-pair alignment shapes the latent space. This characterization theoretically reveals how topological constraints mitigate class confusion driven by semantic drift.

- We derive rigorous excess-risk bounds for both unsupervised pre-training and supervised downstream tasks. Crucially, our downstream bound explicitly incorporates the effective number of connected neighbors, quantitatively proving that richer topological connectivity enhances generalization performance.

- We establish a measure-theoretic framework to unify the analysis of augmentation strengths. We prove—and empirically validate—that our approach effectively alleviates the reliance of contrastive learning on specific data augmentation strategies.

## 2. Related Work

### 2.1. Theoretical Understanding of Contrastive Learning

Initial interpretations of InfoNCE as Mutual Information maximization (Oord et al., 2018; Hjelm et al., 2018; Tian et al., 2020; Tosh et al., 2021; Tsai et al., 2021) have been challenged due to the poor bias-variance trade-offs of variational estimators and unverifiable assumptions (Song & Ermon, 2019; Chuang et al., 2020). Consequently, the focus shifted to geometric principles of alignment and uniformity (Wang & Isola, 2020), though this perspective often overlooks the complex interplay between augmentations and latent topology (Wang et al., 2022). Besides, the behavior of InfoNCE is also studied from the sparse coding model (Wen & Li, 2021), the expansion assumption (Wei et al., 2020), stochastic neighbor embedding (Hu et al., 2022), and augmentation robustness (Zhao et al., 2023). Regarding generalization, existing guarantees typically rely on impractical assumptions, such as the conditional independence of positive pairs (Saunshi et al., 2019; Lee et al., 2021) or negligible intra-class support (Huang et al., 2021). Furthermore, While spectral approaches (HaoChen et al., 2021) have analyzed graph connectivity, but it is only applicable to their own spectral contrastive loss. Crucially, they lack a mechanism to explicitly regulate the topological connectivity induced by positive pairs, leaving the model vulnerable to the formation of unintended bridges between distinct semantic clusters.

### 2.2. Persistent Homology in Machine Learning

Persistent Homology has emerged as a rigorous framework for inferring robust geometric features from complex (Carlsson, 2009; Edelsbrunner et al., 2002) and increasingly analyzing deep representations (Hofer et al., 2017; Magai & Ayzenberg, 2022; Chazal & Michel, 2021; Tulchinskii et al., 2023; Barannikov et al., 2021; Trofimov et al., 2023). While prior works have utilized Betti numbers or persistence diagrams primarily as diagnostic tools to assess manifold entanglement or generation quality (Watanabe & Yamana, 2022; Reininghaus et al., 2015), efforts to apply differentiable Persistent Homology (PH) for regularization have

typically focused on preserving the topology of the input space or simplifying complexity (Moor et al., 2020; Hofer et al., 2019). To our knowledge, our work is the first to employ PH to actively sculpt the connectivity of the latent contrastive space.

# 3. Topology-Aware Contrastive Learning

We first identify the geometric limitations of standard contrastive learning (CL) and then introduce a topological alternative leveraging Persistent Homology to explicitly regulate latent space connectivity.

## 3.1. Limitations of Standard Contrastive Learning

We consider the standard contrastive learning framework where an encoder $f : \mathcal{X} \to \mathbb{R}^d$ maps input data to a normalized representation space. For a sample $x$, a positive pair $(x, x^+)$ is generated via stochastic augmentations, while negative samples $\{x_i^-\}_{i=1}^m$ are drawn independently. The encoder is trained by minimizing the InfoNCE loss:

$$\mathcal{L}_{\text{NCE}}(f) = \mathbb{E}_{p(x,x^+)}\mathbb{E}_{\{p(x_i^-)\}}$$
$$\left[ -\log \frac{\exp\left(f(x)^\top f(x^+)\right)}{\sum_{i=1}^m \exp\left(f(x)^\top f(x_i^-)\right)} \right], \quad (1)$$

where $p(x)$ is the data distribution, and $p(x, x^+)$ is the joint distribution of positive pairs.

Implicitly, Eq. (1) operates on a strict geometric assumption: invariance is learned by minimizing the metric distance between positive pairs, i.e., $d(z, z^+) \to 0$. While effective for instance discrimination, this "zero-distance" objective forces the encoder to collapse local variations toward a singularity. Excessive geometric contraction has been observed to distort the intrinsic topological structure of data particularly when strong augmentations introduce semantic ambiguity (Huang et al., 2021). This topology-agnostic contraction can induce unintended inter-class connections, leading to class confusion. To mitigate this, we propose shifting the objective from strict geometric collapse to topological connectivity—ensuring positive pairs remain within the same connected component of the latent structure without collapsing to a single point.

## 3.2. Topological Perspective: Persistent Homology

To quantify and regulate this connectivity, we adopt PH (Edelsbrunner & Harer, 2010). We focus specifically on 0-dimensional homology ($H_0$), which characterizes the evolution of connected components across spatial scales.

**The Vietoris-Rips Filtration.** Let $S = \{z_1, \ldots, z_b\} \subset \mathbb{R}^d$ be a mini-batch of latent representations. We analyze the topology of $S$ via the Vietoris-Rips (VR) complex. The 1-skeleton of the VR complex $\mathcal{V}_\epsilon(S)$ at scale $\epsilon$ is a graph containing all edges $(z_i, z_j)$ such that $d(z_i, z_j) \leq \epsilon$. Varying $\epsilon$ yields a filtration:

$$\emptyset \subseteq \mathcal{V}_{\epsilon_0}(S) \subseteq \mathcal{V}_{\epsilon_1}(S) \subseteq \cdots \subseteq \mathcal{V}_{\epsilon_M}(S), \quad (2)$$

where $0 = \epsilon_0 < \epsilon_1 < \cdots < \epsilon_M$ are the unique pairwise distances in $S$ (see Figure 1(c)).

**0-Dimensional Persistence and Death Times.** As the scale $\epsilon$ increases, distinct components in the graph merge. A merging event at scale $\epsilon_t$ signifies the "death" of a connected component that persisted from $\epsilon = 0$. The collection of these critical scales forms the multi-set of death times, denoted as $\mathfrak{d}(S)$. Crucially, each $t \in \mathfrak{d}(S)$ corresponds to a critical threshold $\epsilon_t$ defined exactly by the length of the specific edge $(z_i, z_j)$ that triggers the merger.

**Topological Hypothesis.** In this framework, standard contrastive learning is topologically equivalent to forcing specific death times to zero. Instead of indiscriminate minimization, we propose to explicitly regulate the distribution of $\mathfrak{d}(S)$. By maintaining death times within a target interval, we ensure representations form persistent topological features rather than collapsing to singularities.

## 3.3. Topology-Aware Connectivity Loss

Building on the topological intuition that representations should form persistent connected components, we introduce a regularization term to explicitly control the distribution of death times.

**The Connectivity Objective.** Following Hofer et al. (2019), we define the connectivity loss $\mathcal{L}_\eta$ that penalizes the deviation of the critical scales $\epsilon_t$—corresponding to the observed death times in $\mathfrak{d}(S)$—from a connectivity radius $\eta$:

$$\mathcal{L}_\eta(S) = \sum_{t \in \mathfrak{d}(S)} |\eta - \epsilon_t|. \quad (3)$$

Minimizing Eq. (3) forces the latent graph $\mathcal{V}_\eta(S)$ to remain connected at scale $\eta$, effectively acting as a structural constraint on the lifespan of topological features.

To operationalize this for contrastive learning, let $S = \{x_1, \ldots, x_b\}$ denote a set of augmented views derived from the same anchor. We aim to control the connectivity of the latent representations $f(S) = \{f(x_1), \ldots, f(x_b)\}$. While Eq. (3) is defined over abstract death times, optimization requires expressing the loss in terms of specific sample pairs. Following the methodology of Hofer et al. (2019), we identify the specific edges responsible for merging connected components using an indicator function $\mathbb{I}_{i,j}$. This function acts as a topological gate, selecting only the critical pairs $(f(x_i), f(x_j))$ whose pairwise distance corresponds to a death time $t \in \mathfrak{d}(f(S))$:

$$\mathbb{I}_{i,j}(f(S)) = \begin{cases} 1, & \exists t \in \mathfrak{d}(f(S)) : \epsilon_t = \|f(x_i) - f(x_j)\|, \\ 0, & \text{otherwise.} \end{cases}$$

The topology-aware loss is then computed as the expected deviation over these critical pairs:

$$\mathcal{L}_\eta(f) = \mathbb{E}_{p(x_i, x_j)} |\eta - \|f(x_i) - f(x_j)\|| \cdot \mathbb{I}_{i,j}(f(S)). \quad (4)$$

This mechanism imposes a bi-directional constraint on the latent structure: it penalizes large distances to ensure connectivity while penalizing overly small distances to maintain separability. Thus, it prevents the trivial collapse of local geometry while ensuring class-wise structural integrity.

**Total Training Objective.** Our final topology-aware contrastive objective combines the semantic discrimination of InfoNCE with the structural preservation of the connectivity loss:

$$\mathcal{L}_{\eta\text{-NCE}}(f) = \mathcal{L}_{\text{NCE}}(f) + \lambda \mathcal{L}_\eta(f), \quad (5)$$

where $\lambda$ controls the regularization strength. By optimizing this joint objective, the encoder learns representations that are semantically discriminative yet topologically robust.

### 3.4. Excess Risk Bound for Pre-training

While the joint objective in Eq. (5) provides a practical mechanism for learning topologically robust representations, it is crucial to establish its theoretical validity regarding learnability. As the discrete sorting operations via the indicator $\mathbb{I}_{i,j}$ in the exact loss (Eq. (5)) impede standard Rademacher complexity analysis, we employ a smooth relaxation, upper-bounding the topological constraint via the expected squared deviation from $\eta$. This yields the following population loss:

$$\mathcal{L}_\eta(f) = \mathbb{E}_{x, x^+} \left[ (\eta - \|f(x) - f(x^+)\|)^2 \right]. \quad (6)$$

In the empirical setting, we consider a finite dataset $\widehat{\mathcal{X}} = \{\bar{x}_1, \ldots, \bar{x}_n\}$ of size $n$, with $\widehat{\mathcal{P}}_\mathcal{X}$ denoting the uniform distribution over $\widehat{\mathcal{X}}$. Positive pairs $(x, x^+)$ are sampled independently from the augmentation distribution $\mathcal{A}(\bar{x})$, while $m$ negative samples $\{x_i^-\}_{i=1}^m$ are drawn from the augmentation distributions $\mathcal{A}(\bar{x}_i)$ of distinct samples $\bar{x}_i \sim \widehat{\mathcal{P}}_\mathcal{X}$, with $\bar{x}_i \neq \bar{x}$. Fixing $\lambda = 1$ to focus on the interplay between contrastive and topological terms, the empirical risk $\widehat{\mathcal{L}}_{\text{NCE}}(f)$ and $\widehat{\mathcal{L}}_{\eta\text{-NCE}}(f)$ is defined as:

$$\widehat{\mathcal{L}}_{\text{NCE}}(f) := \mathbb{E}_{\substack{\bar{x} \sim \widehat{\mathcal{P}}_\mathcal{X} \\ x \sim \mathcal{A}(\bar{x}), \ x^+ \sim \mathcal{A}(\bar{x})}} \mathbb{E}_{\substack{\{\bar{x}_i\}_{i=1}^m \sim \widehat{\mathcal{P}}_\mathcal{X} \\ x_{1:m}^- \sim \mathcal{A}(\bar{x}_{1:m})}} \\ \left[ -\log \frac{\exp(f(x)^\top f(x^+))}{\sum_{i=1}^m \exp(f(x)^\top f(x_i^-))} \right], \quad (7)$$

$$\widehat{\mathcal{L}}_{\eta\text{-NCE}}(f) = \widehat{\mathcal{L}}_{\text{NCE}}(f) + \\ \mathbb{E}_{\substack{\bar{x} \sim \widehat{\mathcal{P}}_\mathcal{X} \\ x \sim \mathcal{A}(\bar{x}), \ x^+ \sim \mathcal{A}(\bar{x})}} (\eta - \|f(x) - f(x^+)\|)^2. \quad (8)$$

We utilize Rademacher complexity to measure the capacity of the hypothesis class $\mathcal{F}$. The following theorem characterizes the excess risk between the learned empirical minimizer $\hat{f}$ and the optimal population minimizer $f^*$.

**Theorem 3.1.** *Let $\mathcal{F}$ be a hypothesis class of functions $f : \mathcal{X} \to \mathbb{R}^k$ such that $\|f(x)\|_\infty \leq \gamma$ for all $x$. Let $f^* \in \mathcal{F}$ be a minimizer of the population loss $\mathcal{L}_{\eta\text{-NCE}}$, and $\hat{f} \in \mathcal{F}$ be a minimizer of the empirical loss $\widehat{\mathcal{L}}_{\eta\text{-NCE}}$. Assuming the logarithmic loss component is $\ell$-Lipschitz, with probability at least $1 - \delta$, we have:*

$$\mathcal{L}_{\eta\text{-NCE}}(\hat{f}) \leq \mathcal{L}_{\eta\text{-NCE}}(f^*) + c_1 \cdot \widehat{\mathcal{R}}_{n/(m+1)}(\mathcal{F}) \\ + c_2 \cdot \left( \sqrt{\frac{2(m+1)\log(2/\delta)}{n}} + \frac{\delta}{2} \right). \quad (9)$$

*where $\widehat{\mathcal{R}}(\cdot)$ denotes the empirical Rademacher complexity, $c_1 \lesssim (m\ell\gamma + \gamma + \eta)k$, and $c_2 \lesssim k\gamma^2 + \eta^2$.*

**Remark**: Standard learning theory results establish that $\widehat{\mathcal{R}}_n(\mathcal{F}) = \mathcal{O}(1/\sqrt{n})$ (Bartlett & Mendelson, 2002). Theorem 3.1 demonstrates that while topological constraints introduce a controlled overhead via $\eta$-dependent constants $(c_1, c_2)$, they crucially preserve this $\mathcal{O}(1/\sqrt{n})$ convergence rate. Specifically, for $\eta, \ell, \gamma = \mathcal{O}(1)$, the sample complexity scales as $\mathcal{O}(m^2 k^2/\epsilon^2)$, confirming that the objective is theoretically learnable and empirical minimization yields near-optimal population risk.

## 4. Topological Analysis of CL

### 4.1. The Mechanism of Connectivity

Contrastive learning induces latent clustering via implicit connectivity. Due to the competing optimization objectives—standard semantic alignment versus topological regularization—we can only expect the connectivity scale to lie in an interval $[\alpha, \beta]$ around $\eta$. This behavior is captured in the following definition.

**Definition 4.1.** ($\alpha$–$\beta$ Connectivity). Let $\{C_k\}_{k=1}^K$ be latent semantic classes and $\bar{x} \in \bigcup_{k=1}^K C_k$. For a set of augmented views $A(\bar{x})$, we define a finite set $\alpha$–$\beta$ connected set as:

$$S = \left\{ \forall x, x^+ \in A(\bar{x}) \ \middle| \ \alpha \leq \|f(x) - f(x^+)\| \leq \beta \right\}.$$

We say $S$ is $\alpha$–$\beta$ connected. This definition captures the ideal state of representation learning: pairs are neither too close nor too far. However, since the loss only explicitly controls positive pairs, we require an assumption on the underlying data distribution to guarantee that this pairwise control propagates to class-level connectivity.

**Assumption 4.2.** (Latent Semantic Continuity). For any $\bar{x}, \bar{x}_1 \in C_k$, we assume their augmented views $\{A(\bar{x}) \cup A(\bar{x}_1)\}$ contain a subset of samples that form an $\alpha$–$\beta$ connected path in the feature space.

This assumption is consistent with established perspectives (HaoChen et al., 2021; Wang et al., 2022), reflecting the intuition that while pixel-level variations may be large, intra-class samples share invariant semantic content that bridges them in the representation space. Based on this, we define an augmented-annular notion of neighborhood.

**Definition 4.3.** (Augmented-Annular Neighborhood). For a sample $\bar{x} \in C_k$, a sample $\bar{x}_1 \in C_k \setminus \{\bar{x}\}$ is an augmented-annular neighbor of $x$ if there exist augmentations $x \in A_n(\bar{x})$ and $x_1 \in A_n(\bar{x}_1)$ such that $\|f(x) - f(x_1)\|$ satisfy $\alpha-\beta$ connected. The set of all such intra-class neighbors is denoted as $B_n(\bar{x}; \alpha, \beta)$, and its cardinality, the augmented-annular degree, is denoted by $\deg_n(\bar{x})$.

Intuitively, $\deg_n(\bar{x})$ measures the number of intra-class samples effectively linked to $x$ through the topological constraints of the loss function. The following theorem establishes a lower bound on this connectivity.

**Theorem 4.4.** (Connectivity Lower Bound). Let $A_p(x)$ be the set of $p$ augmentations for a sample $x$. Given subset $S_q \subset A_p(x)$ of size $q$ ($2 \leq q \leq p$) is $\alpha-\beta$ connected. Building on Assumption 4.2, for any $x \in C_k$, there exists a subset of neighbors $B_p(x; \alpha, \beta) \subseteq B_n(x; \alpha, \beta)$ such that:

$$\deg_p(x) \geq p - q + 1. \quad (10)$$

**Remark**: Detailed proof is provided in the Appendix C. Theorem 4.4 establishes a quantitative lower bound of $p - q + 1$ on the augmented-annular degree, guaranteeing that each sample maintains effective linkages to at least this many intra-class neighbors. This result illuminates the trade-off between augmentation richness($p$) and topological strictness($q$): increasing $p$ enhances the richness of augmentations to expand the pool of potential connections, while decreasing the parameter $q$ relaxes the structural requirement to admit "milder" connections(i.e., not requiring strong $\alpha-\beta$ connectivity), thereby maximizing the effective neighbor count. Overall, the augmented-annular degree provides a minimum guarantee on the number of effective neighbors, which in turn ensures that samples from the same class form connected clusters in the representation space.

### 4.2. The Mechanism of Separability and Confusion

Complementing our connectivity analysis, we now examine the role of separability in shaping learning behavior. We first quantify the capacity of the representation space under topological constraints. The following lemma provides an upper bound on the maximum number of samples that can be accommodated within the target annulus while maintaining a minimum separation $\delta$.

**Lemma 4.5.** (Capacity of the Augmented Annulus). For $\bar{x} \in C_k$, let $\mathcal{N} = \{x \in A_n(\bar{x}) | \alpha \leq \|f(x)\| \leq \beta\}$. Set $\mathcal{M}(\mathcal{N}, \delta) = \sup \{|S| : S \subset N, \forall x, x^+ \in S, \|f(x) -$

$f(x^+)\| \geq \delta\}$. *Then we have:*

$$\mathcal{M}(\mathcal{N}, \delta) \leq \left\lceil \frac{\beta - \alpha}{\delta} \right\rceil \cdot \exp\left( nR(2 \arcsin \frac{\delta}{2\beta}) \right). \quad (11)$$

This bound, derived from spherical coding theory (Kabatian-sky & Levenshtein, 1978; Cohn & Zhao, 2014), is strictly tighter than volume-based approximations. It reveals that separability is governed by the interplay between the shell width $\beta - \alpha$ and the separation $\delta$. In our framework, setting $\delta = \eta$ ensures that the space is not overcrowded, preserving the distinctness of features. In what follows, we discuss how separability takes effect.

**Theorem 4.6.** (Topology-Agnostic Confusion). *Let* $B_k = \{\bar{x}, \bar{x}_1 \in C_k, \|f(\bar{x}) - f(\bar{x}_1)\| < \varepsilon\}$ *with* $\varepsilon \in (0, \alpha]$ *denote the intra-class collapse event, the inter-class connectivity event as* $D_k = \{\bar{y} \in C_{\neg k}, \bar{x} \in C_k : \bar{y}$ *is the augmented-annular neighbor of* $\bar{x}\}$ *and* $W$ *denote the class confusion event. When* $\mathcal{M} > \left\lceil \frac{\beta - \alpha}{\alpha} \right\rceil \cdot \exp\left( nR(2 \arcsin \frac{\alpha}{2\beta}) \right)$ *and given* $P(D_k | B_k) = \rho$, *we have:*

$$P(W) \geq \rho \cdot \sum_{k=1}^{K} P(B_k), \quad (12)$$

*where* $P(\cdot)$ *denotes the probability of the corresponding event.*

**Remark:** Theorem 4.6 provides a formal explanation for the "topology-agnostic collapse" phenomenon. The inequality implies that the probability of class confusion is strictly lower-bounded by the degree of feature collapse $P(B_k)$. Consider the motivating example: if an augmentation of a "car" (cropped to a wheel) and a "motorcycle" (cropped to a wheel) are both forced to collapse to their respective class centroids ($P(B_k)$ is high), the semantic overlap of the "wheel" features acts as a bridge, pulling the two distinct manifolds together ($\rho$ is high). By enforcing $\alpha-\beta$ connectivity, our method explicitly minimizes $P(B_k)$ by prohibiting zero-distance mappings. This lowers the theoretical floor for class confusion, allowing the model to maintain separation between classes even in the presence of ambiguous local features.

### 4.3. Downstream Generalization Bounds

Having quantified the risk of class confusion under feature collapse, we now address the ultimate goal: **linking pre-trained topological integrity to downstream classification performance**. In this section, we derive a generalization bound that explicitly incorporates the effective number of connected neighbors ($C$), bridging the gap between our unsupervised topological objective and supervised risk.

We use the class-wise mean representation $\mu_k = \mathbb{E}_{p(x|y=k)}[f(x)]$ as the weight $w_k$ of the classifier $g$. The

mean Cross-Entropy loss is defined as:

$$\mathcal{L}_{\text{CE}}^{\mu}(f) = \mathbb{E}_{p(x,y)} \left[ - \log \frac{\exp(f(x)^{\top}\mu_y)}{\sum_{i=1}^{K} \exp(f(x)^{\top}\mu_i)} \right] \quad (13)$$

Crucially, it strictly upper-bounds the optimal linear classification risk, i.e., $\mathcal{L}_{\text{CE}}^{\mu}(f) \geq \min_g \mathcal{L}_{\text{CE}}(f,g)$, and has been shown to achieve performance comparable to learned weights (Saunshi et al., 2019).

**Theorem 4.7.** *(Connectivity-Dependent Generalization). For any $f \in F$ with $|C_k| = N$, its downstream classification risk $\mathcal{L}_{\text{CE}}^{\mu}(f)$ can be bounded by the contrastive learning risk $\mathcal{L}_{\text{NCE}}(f)$ and the connectivity risk $\mathcal{L}_{\eta}(f)$:*

$$\begin{aligned} \mathcal{L}_{\text{CE}}^{\mu}(f) + \log(m/K) \; \leq \; & \mathcal{L}_{\text{NCE}}(f) + b \cdot \sqrt{\mathcal{L}_{\eta}(f)} \\ & + \mathcal{O}\left( \sqrt{N/C} + m^{-1/2} \right), \end{aligned} \quad (14)$$

*where $m$ is the number of negative samples, and $C \geq p - q + 1$ (from Theorem 4.4) denotes the number of mutually connected intra-class samples. The coefficient $b = \sqrt{\frac{1}{2(1-\omega)}}$ is determined by the positive pairs samples correlation $\omega = \text{tr}(\Sigma_{12})/\text{tr}(\Sigma)$.*

**Remark:** In contrast to existing bounds that depend solely on global intra-class variance (Saunshi et al., 2019; Wang et al., 2022), Theorem 4.7 explicitly incorporates topological terms—the connectivity loss $\mathcal{L}_{\eta}$ and neighbor count $C$—offering the following key advantages:

- **Decay via Connectivity:** The error term scales inversely with $C$, formally proves that generalization error decays as the number of connected neighbors increases. Since $C$ scales with augmentation richness ($p$), this guarantees that improving local connectivity directly enhances downstream performance.

- **Active Variance Control:** Unlike standard bounds that rely on implicit variance reduction, minimizing $\mathcal{L}_{\eta}$ actively constrains the variance to a controllable range, tightening the upper bound on the risk.

- **The Correlation-Connectivity Trade-off:** Our bound reveals a fundamental balancing mechanism similar to margin theory, Specifically, in the weak augmentation regime ($\omega \to 1$), the coefficient $b$ diverges to infinity, causing the bound to degenerate; this confirms that sufficient augmentation strength is requisite for generalization. Conversely, in the overly strong augmentation regime ($\omega \to 0$), while $b$ stabilizes, the connectivity count $C$ may diminish due to broken links, increasing the error term $\mathcal{O}(C^{-1})$. Thus, optimal generalization is achieved not at the extremes, but by balancing pairwise correlation ($\omega$) with topological density ($C$).

**Limitations.** We acknowledge that the separation constraint ($\eta > 0$) prevents the intra-class variance from vanishing to zero. However, this is a deliberate trade-off: by preventing variance from reaching zero, we avoid the catastrophic class confusion proved in Theorem 4.6. Simultaneously, this also mitigates the heavy reliance on data augmentation; we analyze this behavior theoretically in Section 5 and verify it empirically in Section 6.

# 5. Mitigating Augmentation Sensitivity

While the generalization guarantees in Section 4 rely on topological integrity, practical contrastive learning often hinges delicately on augmentation design. Standard frameworks face a dilemma: weak augmentations induce representation collapse ($R_\epsilon$), while excessively strong augmentations violate semantic consistency ($R_\delta$). In this section, we analyze how $\alpha$–$\beta$ connectivity mitigates this sensitivity. We introduce a measure-theoretic framework to unify discrete and continuous augmentations, deriving a bound on augmentation-induced errors.

## 5.1. Augmentation Error and Soft Constraints

We formally categorize the failure modes of augmentation strategies into two distinct types: *strong augmentation error* and *weak augmentation error*.

The **Strong Augmentation Error** $R_\delta$ occurs when the augmentation distance exceeds a semantic threshold $\delta \gg \beta$, indicating a loss of semantic integrity:

$$R_\delta = \Pr \left[ x \in \bigcup_{k=1}^{K} C_k : \sup_{x_1, x_2 \in A(x)} \|f(x_1) - f(x_2)\| > \delta \right]. \quad (15)$$

The **Weak Augmentation Error** $R_\epsilon$ occurs when the distance falls below a collapse threshold $\epsilon \ll \alpha$, indicating indistinguishable representations:

$$R_\epsilon = \Pr \left[ x \in \bigcup_{k=1}^{K} C_k : \inf_{x_1, x_2 \in A(x)} \|f(x_1) - f(x_2)\| < \epsilon \right]. \quad (16)$$

The total augmentation error probability is defined as $R = R_\delta \cup R_\epsilon$.

To analyze the relationship between the loss function and these errors, we consider the soft-constraint formulation of our topological objective:

$$\begin{aligned} \mathcal{L}_{\alpha-\beta}(f) = \mathbb{E}_{x \in C_k} \mathbb{E}_{x_1, x_2 \in A(x)} [ & (\alpha - \|f(x_1) - f(x_2)\|)_+ \\ & + (\|f(x_1) - f(x_2)\| - \beta)_+], \end{aligned} \quad (17)$$

where $(u)_+ = \max(u, 0)$. This term penalizes distances outside the $[\alpha, \beta]$ interval, acting as a regularizer that prevents the model from blind over-contraction or semantic disruption.

## 5.2. Theoretical Guarantee

To provide a principled characterization that unifies discrete operations (e.g., cropping, flipping) and continuous transformations (e.g., color jittering), we formulate the augmentation process as a measure space.

**Definition 5.1.** (Augmentation Measure Space). We define the tuple $(\mathcal{A}_x, \Sigma_x, \mu_x)$, where $\mathcal{A}_x$ is the set of all possible augmented samples of $x$, $\Sigma_x$ is a $\sigma$-algebra over $\mathcal{A}_x$, and $\mu_x$ is a mixture probability measure:

$$\mu_x = \lambda_d \mu_x^{(d)} + \lambda_c \mu_x^{(c)}, \quad \text{with } \lambda_d + \lambda_c = 1. \quad (18)$$

Furthermore, we define

$$\mu_x^{(d)} = \frac{1}{m} \sum_{\gamma=1}^{m} \varphi_{A_\gamma(x)}, \quad \mu_x^{(c)} = \int_{\theta \in \Theta} p_{A_\theta(x)} \, d\theta, \quad (19)$$

where $\{A_\gamma(\cdot) \colon \gamma \in [m]\}$ represents the set of discrete transformations, and $\{A_\theta : \theta \in \Theta \subseteq [0,1]^n\}$ represents the continuous family parameterized by $\theta$ with density $p_{A_\theta(x)}$. Here, $\varphi_{A_\gamma(x)}$ denotes the Dirac measure.

We assume that $A_\theta(x)$ varies smoothly with respect to the augmentation parameter $\theta$, satisfying the $L$-Lipschitz condition: $\|A_{\theta_1}(x) - A_{\theta_2}(x)\| \leq L \|\theta_1 - \theta_2\|, \forall x, \theta_1, \theta_2$. Then, the following theorem establishes that optimizing the topological soft constraint $\mathcal{L}_{\alpha-\beta}$ directly minimizes the upper bound of the total augmentation error $R$.

**Theorem 5.2.** *(Bound on Augmentation Error). Given an encoder $f$ and a valid smoothing parameter $h \in (0, \min\{\frac{\sqrt{n}L\kappa(2\alpha^3 - 3\beta^2\varepsilon)}{6\beta^2}, \frac{\delta - \beta \max\{4m^2, 2mm', 4m'^2\}}{2\sqrt{n}L\kappa}\})$, the total probability of error caused by overly strong or weak augmentations is bounded by:*

$$R \leq \mathcal{O}(\delta, \varepsilon) \cdot \mathcal{L}_{\alpha-\beta}(f), \quad (20)$$

*where the coefficient $\mathcal{O}(\delta, \epsilon)$ is defined as:*

$$\mathcal{O}(\delta, \varepsilon) = \inf_h \frac{\alpha^2}{\alpha^3 - 3\beta^2(\varepsilon/2 + \sqrt{n}Lh\kappa)} + \frac{\max\{4m^2, 2mm', 4m'^2\}}{\delta - 2\sqrt{n}Lh\kappa - \beta \max\{4m^2, 2mm', 4m'^2\}}. \quad (21)$$

**Remark:** Theorem 5.2 implies that a small topological loss $\mathcal{L}_{\alpha-\beta}(f)$ is a sufficient condition for a low augmentation error rate. Unlike standard contrastive learning, which is highly sensitive to the precise tuning of augmentation hyperparameters, our framework implies that as long as the loss is minimized, the model strictly bounds the probability of both semantic drift ($R_\delta$) and feature collapse ($R_\epsilon$). This theoretically confirms that enforcing $\alpha-\beta$ connectivity actively effectively buffers the model against suboptimal augmentation strategies. We provide a detailed empirical verification in Section 6.

## 6. Experiments

In this section, we empirically validate the theoretical claims made in Sections 4 and 5. Our evaluation focuses on three key aspects: (1) the improvement in representation quality and topological structure, (2) the sensitivity to augmentation strength, and (3) Impact of Semantic Granularity.

### 6.1. Topological Quality and Performance Analysis

**Setup and Evaluation.** We conduct experiments on CIFAR-10 and CIFAR-100 datasets using a ResNet-50 backbone. Models are pre-trained for 100 epochs with a batch size of 256, utilizing an SGD optimizer with a learning rate of $10^{-3}$ and weight decay of $10^{-6}$. Regarding the structural hyperparameters of our proposed framework, we set the target connectivity radius to $\eta = 0.5$ and generate $n = 4$ positive views for each anchor to ensure sufficient topological density. The data augmentation pipeline follows the standard SimCLR protocol (Chen et al., 2020), including random cropping, color jittering, and horizontal flipping. For evaluation, we employ two primary metrics: (1) Linear Evaluation, where we freeze the pre-trained encoder and fine-tune a linear classifier using Cross-Entropy loss; and (2) k-NN Purity, introduced in Section 4.1 to quantify topological connectivity. For a learned representation $z$, we calculate the proportion of neighbors sharing the same class label among its $k$ nearest neighbors ($k \in \{3, 8, 20\}$).

**Results and Analysis.** Visualizations of the latent space (Figure 2) reveal a sharp contrast in topological structure. As shown in Figure 2(a), the baseline (SimCLR) exhibits severe class confusion within the Vehicles 1 superclass, driven by aggressive geometric contraction. In contrast, Figure 2(b) demonstrates that our topology-aware objective effectively mitigates this confusion; by enforcing $\alpha-\beta$ connectivity, it preserves the distinct separation of intra-class clusters. These visual insights are corroborated by quantitative gains. Table 1 reports that our method consistently outperforms the baseline in $k$-NN Purity across all $k$ values (3, 8, 20) on randomly selected superclasses. On the full CIFAR-100 dataset (Figure 2(d)), these topological constraints translate into a 1.9% improvement in linear accuracy and a 2.63% increase in $k$-NN purity ($k = 3$). Furthermore, Figure 2(c) confirms that regulating connectivity accelerates training convergence. Detailed results for CIFAR-10, alongside comprehensive ablation studies analyzing the impact of key hyperparameters—including the connectivity coefficient $\lambda$, the number of positive pairs, and the target connectivity $\eta$—are provided in Appendix E.

### 6.2. Sensitivity Analysis on Augmentation Strength

**Setup.** To systematically evaluate the sensitivity of learned representations to hyperparameter variations, we isolate two

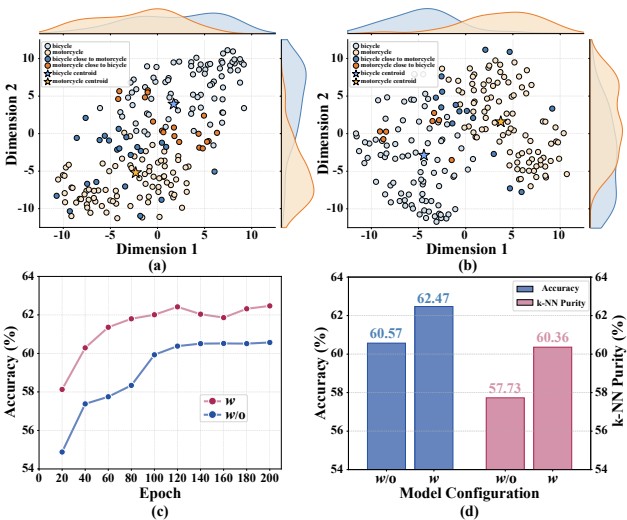

*Figure 2.* Experimental analysis on CIFAR-100. (a)-(b) t-SNE visualizations of the Vehicles 1 superclass (bicycle vs. motorcycle) for the baseline (a) and our method (b). (c) Linear evaluation accuracy curves throughout training epochs. (d) Comparison of final linear accuracy and k-NN purity scores between the baseline (w/o) and our method (w).

critical augmentation components: geometric transformation (via Random Resized Crop) and photometric transformation (via Color Jitter). We define distinct "Weak" and "Strong" regimes to simulate scenarios of insufficient variance and excessive semantic drift, We denote the specific parameter configurations as follows: (a) Weak Geometric: A conservative cropping strategy with a fixed scale of $0.8$ and a narrow aspect ratio interval of $[0.9, 1.1]$, preserving global context. (b) Strong Geometric: An aggressive spatial regularization with a fixed crop scale of $0.2$ and a wide aspect ratio interval of $[0.5, 2.0]$, forcing the model to learn from extremely local views. (c) Weak Photometric: Mild distortion applied with a probability of $0.8$, utilizing intensity factors of $0.1$ for brightness, contrast, and saturation, and $0.02$ for hue. (d) Strong Photometric: Heavy distortion applied with a probability of $1.0$, using intensity factors of $0.8$ for brightness, contrast, and saturation, and $0.2$ for hue. We evaluate the linear evaluation accuracy on CIFAR-100 when these configurations are applied individually and in combination.

**Results and Analysis.** Table 2 demonstrates that our method significantly mitigates the sensitivity to augmentation strength observed in SimCLR. In the Weak Regime, SimCLR succumbs to feature collapse ($R_\epsilon$) due to insufficient variance, reaching only $38.92\%$ Top-1 accuracy. In contrast, our method achieves $43.19\%$ by leveraging the $\eta$-constraint to actively enforce feature separation even without aggressive augmentations. Conversely, in the Strong Regime, our method effectively counters semantic drift ($R_\delta$).

*Table 1.* Comparison of k-NN Purity scores (%) on five randomly selected CIFAR-100 superclasses.

| Superclass | Method | k-NN Purity (%) | | |
|---|---|---|---|---|
| | | $k = 3$ | $k = 8$ | $k = 20$ |
| Vehicles 1 | SimCLR | 64.90 | 62.42 | 59.07 |
| | **Ours** | **67.30** | **64.67** | **60.49** |
| Vehicles 2 | SimCLR | 76.46 | 72.87 | 67.44 |
| | **Ours** | **78.93** | **75.17** | **69.63** |
| Flowers | SimCLR | 54.86 | 52.70 | 48.20 |
| | **Ours** | **58.66** | **55.22** | **49.81** |
| Fish | SimCLR | 64.53 | 61.25 | 54.46 |
| | **Ours** | **67.40** | **62.40** | **56.08** |
| Insects | SimCLR | 56.93 | 53.22 | 48.11 |
| | **Ours** | **60.33** | **55.42** | **49.54** |

*Table 2.* Linear evaluation accuracy (%) on CIFAR-100 under varying augmentation strengths. The configurations (a)-(d) correspond to the Setup in Section 6.2.

| Strength | Augmentation | Top-1 | | Top-5 | |
|---|---|---|---|---|---|
| | | SimCLR | **Ours** | SimCLR | **Ours** |
| Weak | (a) | 32.98 | **35.13** | 61.71 | **64.02** |
| | (c) | 9.1 | **12.56** | 25.03 | **32.18** |
| | (a) + (c) | 38.92 | **43.19** | 67.01 | **71.01** |
| Strong | (b) | 32.03 | **34.84** | 62.16 | **65.15** |
| | (d) | 11.33 | **17.35** | 31.45 | **39.49** |
| | (b) + (d) | 53.70 | **55.72** | 81.38 | **82.64** |

By allowing topological relaxation via the $\eta$-constraint, we accommodate severe distortions, yielding consistent gains in the combined setting where accuracy rises from $53.70\%$ to $55.72\%$. Most notably, under Strong Color (d), our approach achieves a relative gain of over $6\%$, proving its robustness against high semantic ambiguity.

### 6.3. Impact of Semantic Granularity

While our method improves performance on both datasets, we observe that the gains are significantly more pronounced on CIFAR-100 than on CIFAR-10. This disparity highlights the specific mechanism of our contribution: mitigating class confusion. To systematically investigate this, we extend our evaluation to include STL-10 and ImageNet-100, categorizing the datasets into two distinct regimes: the coarse-grained regime (CIFAR-10, STL-10) and the fine-grained regime (CIFAR-100, ImageNet-100). The quantitative results for k-NN purity (k=3) and linear evaluation accuracy are summarized in Table 3.

**The Coarse-Grained Regime:** Datasets in this category are characterized by large intrinsic semantic margins between a limited number of distinct classes (e.g., 10 classes). On CIFAR-10 and STL-10, standard geometric contraction

*Table 3.* Comparison of k-NN purity ($k = 3$) and linear evaluation accuracy (%) across datasets with varying semantic granularities.

| Dataset | k-NN | | Acc | |
|---|---|---|---|---|
| | SimCLR | **Ours** | SimCLR | **Ours** |
| CIFAR-10 | 71.24 | **71.77** | 84.33 | **85.47** |
| STL-10 | 74.59 | **75.27** | 82.47 | **83.21** |
| CIFAR-100 | 57.73 | **60.36** | 60.38 | **62.47** |
| ImageNet-100 | 30.59 | **37.19** | 60.98 | **64.14** |

(SimCLR) already performs reasonably well because the natural distance between classes acts as an inherent buffer against inter-class confusion. Consequently, explicitly regulating topology in this regime yields steady but relatively modest improvements, specifically a $0.53\%$ and $0.68\%$ absolute gain in k-NN purity for CIFAR-10 and STL-10.

**The Fine-Grained Regime:** In stark contrast, fine-grained datasets feature dense semantic spaces with low inter-class variance and complex real-world backgrounds. In such crowded regimes, standard InfoNCE's push for zero-distance alignment forces distinct manifolds to overlap, triggering the topology-agnostic confusion proven in Theorem 4.6. Our topology-aware loss prevents this detrimental over-alignment, yielding substantial performance leaps. Specifically, we observe increases in k-NN purity and accuracy of $2.63\%$ and $2.09\%$ on CIFAR-100, and remarkably, $6.60\%$ and $3.16\%$ on ImageNet-100. By enforcing $\alpha-\beta$ connectivity, our framework preserves the delicate structural boundaries between highly similar classes, confirming that topological regularization is critical when navigating complex, confusable semantic spaces.

## 7. Conclusion

In this work, we address the geometric limitations of standard contrastive learning by proposing a Topology-Aware framework that leverages Persistent Homology to enforce $\alpha - \beta$ connectivity, effectively preventing feature collapse and semantic entanglement. Our theoretical analysis proves that regulating topological connectivity explicitly lowers the lower bound of class confusion probability and tightens generalization risk, while empirical results confirm superior representation quality and significant robustness against augmentation sensitivity. By bridging computational topology and self-supervised learning, this framework offers a principled perspective on learning representations that respect the intrinsic structural complexity of data.

## Acknowledgements

This work was supported by the Major Program of National Natural Science Foundation of China (No. T2495251), the Key Program of the National Natural Science Foundation of China (No. 62136005), and the Special Fund for Science and Technology Innovation Teams of Shanxi Province (No. 202304051001001).

## Impact Statement

This paper presents work whose goal is to advance the field of Machine Learning. There are many potential societal consequences of our work, none which we feel must be specifically highlighted here.

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

# A. Extended Discussion

## A.1. Detailed Comparison with Related Work

To further contextualize our Topology-Aware Contrastive Learning (TACL) framework, we provide a detailed comparison with several recent advancements that intersect with topological or geometric representation learning.

**Comparison with Topological Graph Contrastive Learning.** Recent works such as those by Luo et al. (2023) and TopoGCL (Chen et al., 2024) introduce topological concepts primarily for graph representation learning. These methods focus on extracting topological features directly from structured graph inputs. In contrast, our framework operates on unstructured data (such as images), where the input-level topology is heavily obscured by pixel-level noise. Rather than relying on structural priors from the input space, we dynamically apply Persistent Homology to the Euclidean latent space. Furthermore, while TopoGCL uses Persistent Homology to directly extract topological feature representations, TACL utilizes it strictly as a regulatory mechanism. Specifically, we compute the Vietoris-Rips filtration on mini-batches of augmented views to identify structurally critical sample pairs (i.e., critical edges) and regulate how the model constructs its representations.

**Comparison with Hyperbolic Contrastive Learning.** Standard Euclidean contrastive learning has also been modified by altering the underlying geometry, such as mapping features to hyperbolic space to better capture hierarchical, tree-like relationships (Ge et al., 2023). Unlike this approach, we do not change the base geometry from Euclidean to hyperbolic; instead, we impose an explicit topological constraint within the standard Euclidean space. While hyperbolic embedding aims to preserve an assumed hierarchical taxonomy, our objective is to actively regulate 0-dimensional homology ($H_0$) to manage connected components. This allows us to specifically address the topology-agnostic confusion caused by augmentation overlap, ensuring that distinct manifolds do not entangle when aggressive data augmentations are applied.

**Comparison with Topology-Preserving Representations.** Another line of research focuses on learning topology-preserving representations by matching the latent persistence diagrams with the input manifold (Trofimov et al., 2023). Our method fundamentally diverges from this objective. For visual classification tasks, preserving the input pixel-space topology is often suboptimal, as it lacks high-level semantic meaning. Instead of preserving the existing topology, TACL actively sculpts a new latent topology. By regulating the death times of augmented views toward a target radius $\eta$, we enforce a specific $\alpha–\beta$ connectivity. This directly addresses a critical vulnerability of the standard InfoNCE objective: it prevents the zero-distance feature collapse that allows semantic drift to build spurious bridges between distinct semantic classes.

## A.2. Further Discussions on Theoretical Mechanisms and Modality

**Connectivity Risk vs. Labeling Error.** While the concepts of labeling error explored in prior theoretical literature (HaoChen et al., 2021; Wang et al., 2022) and our proposed connectivity risk are closely related, they represent fundamentally different mechanisms within the context of contrastive learning. Previous theoretical works typically treat labeling error as an inherent, static property of the data augmentation distribution in the input space—an unavoidable noise factor by which the model is passively affected. Conversely, our connectivity risk $\mathcal{L}_\eta$ acts as an active structural constraint within the learned latent space, directly penalizing pairwise distance deviations from a target radius $\eta$.

Furthermore, Section 5.1 formalizes this traditional labeling error concept as the strong augmentation error $R_\delta$. Rather than treating it merely as an inevitable degradation factor in downstream generalization bounds, Theorem 5.2 proves that $R \le \mathcal{O}(\delta, \epsilon) \cdot \mathcal{L}_{\alpha-\beta}(f)$. This demonstrates that explicitly minimizing the topological connectivity risk actively and strictly bounds the total probability of these augmentation-induced errors.

**Cross-Modality Potential.** Although the initial scientific insight of our work—the observation of topology-agnostic confusion—was rooted in analyzing image data behavior under aggressive augmentations, and consequently, our current empirical evaluation focuses on the 2D image domain, the proposed theoretical framework is inherently modality-agnostic. The computation of the Vietoris-Rips filtration relies solely on calculating pairwise distances within a mini-batch of latent representations. This mathematical formulation implies that any data modality relying on data augmentation to generate positive pairs (e.g., time-series, audio, or text data) can directly incorporate our connectivity regularizer. We explicitly highlight this theoretical cross-modality potential, as our topological framework can seamlessly generalize to regulate latent structures and mitigate semantic drift in domains well beyond computer vision.

# B. Proof of Section 3

In this section, we provide the proof details of the theorem in the Section 3.

## B.1. Proof of Theorem 3.1

**Theorem B.1.** *(Recap of theorem 3.1). Let $\mathcal{F}$ be a hypothesis class of functions $f : \mathcal{X} \to \mathbb{R}^k$ such that $\|f(x)\|_\infty \leq \gamma$ for all $x$. Let $f^* \in \mathcal{F}$ be a minimizer of the population loss $\mathcal{L}_{\eta\text{-NCE}}$, and $\hat{f} \in \mathcal{F}$ be a minimizer of the empirical loss $\widehat{\mathcal{L}}_{\eta\text{-NCE}}$. Assuming the logarithmic loss component is $\ell$-Lipschitz, with probability at least $1 - \delta$, we have:*

$$\mathcal{L}_{\eta\text{-NCE}}(\hat{f}) \leq \mathcal{L}_{\eta\text{-NCE}}(f^*) + c_1 \cdot \widehat{\mathcal{R}}_{n/(m+1)}(\mathcal{F}) + c_2 \cdot \left( \sqrt{\frac{2(m+1)\log(2/\delta)}{n}} + \frac{\delta}{2} \right). \tag{22}$$

*where $\widehat{\mathcal{R}}(\cdot)$ denotes the empirical Rademacher complexity, $c_1 \lesssim (m\ell\gamma + \gamma + \eta)k$, and $c_2 \lesssim k\gamma^2 + \eta^2$.*

To complete the proof of Theorem B.1, we first introduce the following definitions and lemmas.

**Definition B.2.** (Empirical loss) Given a dataset $\widehat{\mathcal{X}} = \{\bar{x}_1, \ldots, \bar{x}_n\}$ of size $n$, with $\widehat{\mathcal{P}}_\mathcal{X}$ denoting the uniform distribution over $\widehat{\mathcal{X}}$. Positive pairs $(x, x^+)$ are sampled independently from the augmentation distribution $\mathcal{A}(\bar{x})$. Let $\{x_i^-\}_{i=1}^m$ be a set of $m$ negative samples, where each $x_i^- \sim \mathcal{A}(\bar{x}_i)$ is drawn from a distinct sample $\bar{x}_i \overset{\text{i.i.d.}}{\sim} \widehat{\mathcal{P}}_\mathcal{X} \setminus \{\bar{x}\}$. Setting $\lambda = 1$, the empirical risk is defined as follows:

$$\widehat{\mathcal{L}}_{\eta\text{-NCE}}(f) := \mathbb{E}_{\substack{\bar{x}\sim\widehat{\mathcal{P}}_\mathcal{X} \\ x\sim\mathcal{A}(\bar{x}),\, x^+\sim\mathcal{A}(\bar{x})}} \mathbb{E}_{\substack{\{\bar{x}_i\}_{i=1}^m\sim\widehat{\mathcal{P}}_\mathcal{X} \\ x_{1:m}^-\sim\mathcal{A}(\bar{x}_{1:m})}} \left[ -\log \frac{\exp\left(f(x)^\top f(x^+)\right)}{\sum_{i=1}^m \exp\left(f(x)^\top f(x_i^-)\right)} + \left(\eta - \|f(x) - f(x^+)\|\right)^2 \right]. \tag{23}$$

By definition, it is easy to see that $\widehat{\mathcal{L}}_{\eta\text{-NCE}}(f)$ is an unbiased estimator of $\mathcal{L}_{\eta\text{-NCE}}(f)$: $\mathbb{E}_{\widehat{\mathcal{X}}}[\widehat{\mathcal{L}}_{\eta\text{-NCE}}(f)] = \mathcal{L}_{\eta\text{-NCE}}(f)$.

To make use of the Radmacher complexity theory, we need to write the empirical loss as the sum of i.i.d. terms, which is achieved by the following sub-sampling scheme:

**Definition B.3.** For a dataset $\widehat{\mathcal{X}}$ of size $n$. Setting the block size $B = m + 1$. we sample a permutation $\sigma : [n] \to [n]$ uniformly at random, then we split the permuted indices into $T = n/B$ consecutive blocks:

$$I_t = \{\pi((t-1)B + 1), \ldots, \pi(tB)\}, \quad t = 1, \ldots, T \tag{24}$$

For each block $I_t$, we sample tuples $S = \left\{ (z_t, z_t^+, z_{t1}^-, \ldots, z_{tm}^-) \right\}_{t=1}^T$ :

$$z_t \sim P(\cdot \mid \bar{x}_{\pi((t-1)B+1)})$$

$$z_t^+ \sim P(\cdot \mid \bar{x}_{\pi((t-1)B+1)})$$

$$z_{tj}^- \sim P(\cdot \mid \bar{x}_{\pi((t-1)B+j+1)}), \quad j = 1, \ldots, m$$

We can define the empirical loss on $S$:

$$\widehat{\mathcal{L}}_S(f) = \frac{1}{T} \sum_{t=1}^T \left[ -\log \frac{\exp(f(z_t)^\top f(z_t^+))}{\sum_{j=1}^m \exp(f(z_t)^\top f(z_{tj}^-))} + (\eta - \| f(z_t) - f(z_t^+) \|)^2 \right] \tag{25}$$

It is easy to see that $\widehat{\mathcal{L}}_S(f)$ is an unbiased estimator of $\widehat{\mathcal{L}}_{\eta-\text{NCE}}(f)$: for given $\widehat{\mathcal{X}}$, if we sample $S$ as above, we have $\mathbb{E}_S[\widehat{\mathcal{L}}_S(f)] = \widehat{\mathcal{L}}_{\eta-\text{NCE}}(f)$.

In order to bound the generalization error of the function class $\mathcal{F}$ on the $\mathcal{L}_{\eta\text{-NCE}}(f)$, we will use the following vector-contraction inequality(Ledoux & Talagrand, 2013).

**Lemma B.4.** *(Talagrand's Lemma) If $\phi : \mathbb{R} \to \mathbb{R}$ is an $L$-Lipschitz continuous function, $\mathcal{F}$ is a function class and $\mathcal{R}(\cdot)$ is Rademacher complexity. Then,*

$$\mathcal{R}(\phi \circ \mathcal{F}) \leq L_\phi \mathcal{R}(\mathcal{F}) \tag{26}$$

The following lemma establishes a connection between the Rademacher complexity of feature extractors and that of the loss function defined over tuples (HaoChen et al., 2021).

**Lemma B.5.** *Let $\mathcal{F}$ be a hypothesis class of feature extractors mapping from $\mathcal{X}$ to $\mathbb{R}^k$. Assume that for all $f \in \mathcal{F}$ and all $x \in \mathcal{X}$, the infinity norm of the output is bounded: $\|f(x)\|_\infty \leq \gamma$. For each $i \in [k]$, define the function with a scalar value $f_i : \mathcal{X} \to \mathbb{R}$ and $f_i(x)$ is the $i$-th coordinate of $f(x)$. Let $\mathcal{F}_i := \{f_i \mid f \in \mathcal{F}\}$ denote the class of induced hypotheses for the coordinate $i$ -th coordinate. For $n \in \mathbb{Z}^+$, define the* maximal possible empirical Rademacher complexity *of $\mathcal{F}_i$ over $n$ data points as*

$$\widehat{\mathcal{R}}_n(\mathcal{F}_i) := \max_{x_1,\ldots,x_n \in \mathcal{X}} \mathbb{E}_\sigma \left[ \sup_{f_i \in \mathcal{F}_i} \left( \frac{1}{n} \sum_{j=1}^n \sigma_j f_i(x_j) \right) \right], \tag{27}$$

*where $x_1, x_2, \ldots, x_n$ are in $\mathcal{X}$ and $\sigma_1, \sigma_2, \ldots, \sigma_n$ are independent Rademacher random variables uniformly distributed over $\{\pm 1\}$. Here, We extend Rademacher complexity to function classes with high-dimensional outputs, and define the Rademacher complexity of $\mathcal{F}$ on $n$ samples as: $\max_{i \in [k]} \widehat{\mathcal{R}}_n(\mathcal{F}_i) = \widehat{\mathcal{R}}_n(\mathcal{F})$. Then the empirical Rademacher complexity on any $n$ tuples $S = \left\{(z_t, z_t^+, z_{t1}^-, \ldots, z_{tm}^-)\right\}_{t=1}^n$ can be bounded by*

$$\mathbb{E}_\sigma \left[ \sup_{f \in \mathcal{F}} \left( \frac{1}{n} \sum_{t=1}^n \sigma_t \left( -f(z_t)^\top f(z_t^+) + \log \sum_{j=1}^m \exp(f(z_t)^\top f(z_{tj}^-)) + (\eta - \| f(z_t) - f(z_t^+) \|)^2 \right) \right) \right]$$

$$\leq (4m\ell\gamma + 4\eta + 12\gamma)k \cdot \max_{i \in [k]} \widehat{\mathcal{R}}_n(\mathcal{F}_i). \tag{28}$$

*Proof.*

$$\mathbb{E}_\sigma \left[ \sup_{f \in \mathcal{F}} \left( \frac{1}{n} \sum_{t=1}^n \sigma_t \left( -f(z_t)^\top f(z_t^+) + \log \sum_{j=1}^m \exp(f(z_t)^\top f(z_{tj}^-)) + (\eta - \| f(z_t) - f(z_t^+) \|)^2 \right) \right) \right]$$

$$\overset{(1)}{\leq} \mathbb{E}_\sigma \left[ \sup_{f \in \mathcal{F}} \left( \frac{1}{n} \sum_{t=1}^n \sigma_t f(z_t)^\top f(z_t^+) \right) \right] + \max_{\substack{z_1,z_2,\ldots,z_n \\ z_1^-,z_2^-,\ldots,z_n^-}} m\ell \mathbb{E}_\sigma \left[ \sup_{f \in \mathcal{F}} \left( \frac{1}{n} \sum_{t=1}^n \sigma_t f(z_t)^\top f(z_t^-) \right) \right]$$

$$+ \mathbb{E}_\sigma \left[ \sup_{f \in \mathcal{F}} \left( \frac{1}{n} \sum_{t=1}^n \sigma_t (\eta^2 - 2\eta \| f(z_t) - f(z_t^+) \|) \right) \right] + \mathbb{E}_\sigma \left[ \sup_{f \in \mathcal{F}} \left( \frac{1}{n} \sum_{t=1}^n \sigma_t \| f(z_t) - f(z_t^+) \|^2 \right) \right]$$

$$\overset{(2)}{\leq} (m\ell + 1) \max_{\substack{z_1,z_2,\ldots,z_n \\ z_1^-,z_2^-,\ldots,z_n^-}} \mathbb{E}_\sigma \left[ \sup_{f \in \mathcal{F}} \left( \frac{1}{n} \sum_{t=1}^n \sigma_t f(z_t)^\top f(z_t^-) \right) \right] + 2\eta \mathbb{E}_\sigma \left[ \sup_{f \in \mathcal{F}} \left( \frac{1}{n} \sum_{t=1}^n \sigma_t \| f(z_t) - f(z_t^+) \| \right) \right]$$

$$+ \mathbb{E}_\sigma \left[ \sup_{f \in \mathcal{F}} \left( \frac{1}{n} \sum_{t=1}^n \sigma_t (f(z_t) - f(z_t^+))^2 \right) \right]$$

$$\leq (m\ell + 1)k \max_{\substack{z_1,z_2,\ldots,z_n \\ z_1^-,z_2^-,\ldots,z_n^- \\ i \in [k]}} \mathbb{E}_\sigma \left[ \sup_{f_i \in \mathcal{F}_i} \left( \frac{1}{n} \sum_{t=1}^n \sigma_t f_i(z_t) f_i(z_t^-) \right) \right] + 2\eta k \max_{i \in [k]} \mathbb{E}_\sigma \left[ \sup_{f_i \in \mathcal{F}_i} \left( \frac{1}{n} \sum_{t=1}^n \sigma_t \left| f_i(z_t) - f_i(z_t^+) \right| \right) \right]$$

$$+ k \max_{i \in [k]} \mathbb{E}_\sigma \left[ \sup_{f_i \in \mathcal{F}_i} \left( \frac{1}{n} \sum_{t=1}^n \sigma_t (f_i(z_t) - f_i(z_t^+))^2 \right) \right]$$

$$\overset{(3)}{\leq} (m\ell+1)k \max_{\substack{z_1,z_2,\ldots,z_n \\ z_1^-,z_2^-,\ldots,z_n^- \\ i\in[k]}} \left\{ \frac{1}{4}\mathbb{E}_\sigma \left[ \sup_{f_i\in\mathcal{F}_i} \left( \frac{1}{n}\sum_{t=1}^n \sigma_t(f_i(z_t)+f_i(z_t^-))^2 \right) \right] + \frac{1}{4}\mathbb{E}_\sigma \left[ \sup_{f_i\in\mathcal{F}_i} \frac{1}{n}\sum_{t=1}^n \sigma_t\left( f_i(z_t)-f_i(z_t^-) \right)^2 \right] \right\}$$

$$+ 2\eta k \max_{i\in[k]} \mathbb{E}_\sigma \left[ \sup_{f_i\in\mathcal{F}_i} \left( \frac{1}{n}\sum_{t=1}^n \sigma_t \left| f_i(z_t)-f_i(z_t^+) \right| \right) \right] + k \max_{i\in[k]} \mathbb{E}_\sigma \left[ \sup_{f_i\in\mathcal{F}_i} \left( \frac{1}{n}\sum_{t=1}^n \sigma_t(f_i(z_t)-f_i(z_t^+))^2 \right) \right]$$

$$\overset{(4)}{\leq} 4(m\ell+1)k\gamma \max_{\substack{z_1,z_2,\ldots,z_n \\ i\in[k]}} \mathbb{E}_\sigma \left[ \sup_{f_i\in\mathcal{F}_i} \left( \frac{1}{n}\sum_{t=1}^n \sigma_t f_i(z_t) \right) \right] + (4\eta+8\gamma)k \max_{i\in[k]} \mathbb{E}_\sigma \left[ \sup_{f_i\in\mathcal{F}_i} \left( \frac{1}{n}\sum_{t=1}^n \sigma_t f_i(z_t) \right) \right]$$

$$= (4m\ell\gamma+4\eta+12\gamma)k \max_{\substack{z_1,z_2,\ldots,z_n \\ i\in[k]}} \mathbb{E}_\sigma \left[ \sup_{f_i\in\mathcal{F}_i} \left( \frac{1}{n}\sum_{t=1}^n \sigma_t f_i(z_t) \right) \right]$$

$$= (4m\ell\gamma+4\eta+12\gamma)k \cdot \max_{i\in[k]} \widehat{\mathcal{R}}_n(\mathcal{F}_i)$$

(1) Notice that the term $\log \sum_{j=1}^m \exp(f(z_t)^\top f(z_{tj}^-)) \in \left[ \log m - k\gamma^2,\ \log m + k\gamma^2 \right]$ is bounded. Hence, we can assume that the logarithmic function is $\ell$-Lipschitz continuous in its domain. This allows us to apply Lemma B.4 and further relax the bound. The second part follows directly from the expansion of the squared term; (2) Since negating the functions in a function class doesn't change its Rademacher complexity and the Rademacher complexity of a constant function class is zero. (3) By applying the AM–GM inequality. (4) By applying Lemma B.4 again. $\qquad\square$

*Proof of Theorem B.1.* We know that $\mathbb{E}_S[\widehat{\mathcal{L}}_S(f)] = \widehat{\mathcal{L}}_{\eta\text{-NCE}}(f)$ and $\mathbb{E}_{\widehat{\mathcal{X}}}[\widehat{\mathcal{L}}_{\eta\text{-NCE}}(f)] = \mathcal{L}_{\eta\text{-NCE}}(f)$. According to Definition B.3, the set $S$ is obtained from $\widehat{\mathcal{X}}$ by reordering and sampling, yielding tuples that contain i.i.d. samples. Therefore, we can leverage Rademacher complexity to establish a uniform convergence bound(). In particular, by Lemma B.4, notice that $-f(z_t)^\top f(z_t^+) + \log \sum_{j=1}^m \exp(f(z_t)^\top f(z_{tj}^-)) + (\eta - \|f(z_t) - f(z_t^+)\|)^2$ always take values in range $[-3k\gamma^2 + \log m, 3k\gamma^2 + \log m + \eta^2]$ and $\max_{i\in[k]} \widehat{\mathcal{R}}_{n/(m+1)}(\mathcal{F}_i) = \widehat{\mathcal{R}}_{n/(m+1)}(\mathcal{F})$, we apply the standard Rademacher generalization bound(Mohri et al., 2018): with probability at least $1 - (\frac{\delta}{2})^2$ over the randomness of $\widehat{\mathcal{X}}$ and $S$, the following holds for any $f \in \mathcal{F}$,

$$\mathcal{L}_{\eta\text{-NCE}}(f) \leq \widehat{\mathcal{L}}_S(f) + (8m\ell\gamma+8\eta+24\gamma)\,k \cdot \widehat{\mathcal{R}}_{n/(m+1)}(\mathcal{F}) + (6k\gamma^2+\eta^2) \cdot \sqrt{\frac{2(m+1)\log(2/\delta)}{n}}. \qquad (29)$$

After removing the randomness of $S$, we can obtain both upper and lower bounds on the generalization error of $\widehat{\mathcal{L}}_{\eta\text{-NCE}}(f)$: with probability at least $1 - \frac{\delta}{2}$ over random $\widehat{\mathcal{X}}$, for any $f \in \mathcal{F}$,

$$\mathcal{L}_{\eta\text{-NCE}}(f) \leq \widehat{\mathcal{L}}_{\eta\text{-NCE}}(f) + (8m\ell\gamma+8\eta+24\gamma)\,k \cdot \widehat{\mathcal{R}}_{n/(m+1)}(\mathcal{F}) + (6k\gamma^2+\eta^2) \cdot \left( \sqrt{\frac{2(m+1)\log(2/\delta)}{n}} + \frac{\delta}{2} \right), \quad (30)$$

$$\mathcal{L}_{\eta\text{-NCE}}(f) \geq \widehat{\mathcal{L}}_{\eta\text{-NCE}}(f) - (8m\ell\gamma+8\eta+24\gamma)\,k \cdot \widehat{\mathcal{R}}_{n/(m+1)}(\mathcal{F}) - (6k\gamma^2+\eta^2) \cdot \left( \sqrt{\frac{2(m+1)\log(2/\delta)}{n}} + \frac{\delta}{2} \right). \quad (31)$$

Let $f^* \in \arg\min_{f\in F} \mathcal{L}_{\eta\text{-NCE}}(f)$ and $\hat{f} \in \arg\min_{f\in F} \widehat{\mathcal{L}}_{\eta\text{-NCE}}(f)$. Combining the upper and lower bounds and we get the excess risk bound: with probability at least $1 - \delta$, we can obtain:

$$\mathcal{L}_{\eta\text{-NCE}}(\hat{f}) \leq \mathcal{L}_{\eta\text{-NCE}}(f^*) + (16m\ell\gamma+16\eta+48\gamma)\,k \cdot \widehat{\mathcal{R}}_{n/(m+1)}(\mathcal{F}) + (12k\gamma^2+2\eta^2) \cdot \left( \sqrt{\frac{2(m+1)\log(2/\delta)}{n}} + \frac{\delta}{2} \right),$$
$$(32)$$

Set $c_1 = (16m\ell\gamma+16\eta+48\gamma)k$ and $c_2 = 12k\gamma^2+2\eta^2$ then we finish the proof. $\qquad\square$

# C. Proof of Section 4

In this section, we provide the proof details of the theorem in the Section 4.

## C.1. Proof of Theorem 4.4

**Theorem C.1.** *(Recap of theorem 4.4). Let $A_p(x)$ be the set of p augmentations for a sample x. Given subset $S_q \subset A_p(x)$ of size q ($2 \leq q \leq p$) is $\alpha$–$\beta$ connected. Building on Assumption 4.2, for any $x \in C_k$, there exists a subset of neighbors $B_p(x; \alpha, \beta) \subseteq B_n(x; \alpha, \beta)$ such that:*

$$\deg_p(x) \geq p - q + 1. \tag{33}$$

*Proof.* For $x \in C_k$, we employ an iterative constructive procedure to construct a sequence of points $\bar{x}_1, \ldots, \bar{x}_{p-q+1} \in C_k$ such that:

$$B_p(x; \alpha, \beta) = \{\bar{x}_1, \ldots, \bar{x}_{p-q+1}\} \subset B_A(x; \alpha, \beta) \tag{34}$$

First, we consider a set $S^{(1)} \subset A_n(x)$ with $x \in S^{(1)}$ and $|S^{(1)}| = q$. It follows from Definition 4.1 that the induced subgraph $G_A[S^{(1)}]$ is connected, implying that the set $S^{(1)}$ is $\alpha$-$\beta$ connected. Since $\deg_{S^{(1)}}(x) = |S^{(1)}| \geq 2$, thus there exists $x_1^+ \in S^{(1)} \setminus \{x\}$ adjacent to $x$ in $G_A$.

If there exists $x_1 \in A_n(\bar{x}_1)$, then, by Assumption 4.2, we are guaranteed to find a path corresponding to their augmented samples. Specifically, there exists a valid connection within the distance set $d(x_1^+, x_1) := \left\{ \|f(x_1^+) - f(x_1)\| \mid x_1^+ \in A(x), \ x_1 \in A(\bar{x}_1) \right\}$ that falls within $[\alpha, \beta]$. Consequently, according to Definition 4.1, we conclude that the associated views $x$ and $\bar{x}_1$ satisfy the $\alpha$-$\beta$ connectivity. Furthermore, it follows from Definition 4.3 that $\bar{x}_1$ is an augmentation-annulus neighbor of $x$. Thus, we know $\bar{x}_1 \in B_A(x, \alpha, \beta)$ and $\deg_A(x) = 1$.

Next, we consider the general step. Given multiple sets $S^{(i)} \subset A_n(x)$ with $x_i^+ \in S^{(i)}$ and $|S^{(i)}| = q$, we can construct $B^{(i)} = \{\bar{x}_1, \bar{x}_2, \cdots, \bar{x}_i\}$, whose elements are mutually distinct and contained in $B_A(x; \alpha, \beta)$. For $0 < i \leq p - q$, the number of remaining points is at least

$$|A_n(x) \setminus \{x, x_1^+, \cdots, x_i^+\}| = p - 1 - i \geq p - 1 - (p - q) = q - 1. \tag{35}$$

Finally, when $i = p - q$, we can select the remaining $q - 1$ points together with $x$ to form the set $S^{(i+1)}$ with $|S^{(i+1)}| = q$. Again, we can select a new point $x_{i+1}^+ \in S^{(i+1)}$, and subsequently, $\bar{x}_{i+1} \in B_A(x; \alpha, \beta)$ can be chosen to be distinct from all previously constructed points. Hence, we can find $p - q + 1$ points, which together constitute the set $B_p(x; \alpha, \beta) = \{\bar{x}_1, \bar{x}_2, \cdots, \bar{x}_{p-q+1}\} \subset B_A(x; \alpha, \beta)$. Once there are no remaining points, the iteration is halted. Then, we finish the proof. $\square$

## C.2. Proof of Lemma 4.5 and Theorem 4.6

**Lemma C.2.** *(Recap of lemma 4.5). For $\bar{x} \in C_k$, let $\mathcal{N} = \{x \in A_n(\bar{x}) | \alpha \leq \|f(x)\| \leq \beta\}$. Set $\mathcal{M}(\mathcal{N}, \delta) = \sup \{|S| : S \subset N, \forall x, x^+ \in S, \|f(x) - f(x^+)\| \geq \delta\}$. Then we have:*

$$\mathcal{M}(\mathcal{N}, \delta) \leq \left\lceil \frac{\beta - \alpha}{\delta} \right\rceil \cdot \exp\left( 2nR \arcsin \frac{\delta}{2\beta} \right). \tag{36}$$

*Proof.* First, We divide the radial interval $[\alpha, \beta]$ into $L = \lceil (\beta - \alpha)/\delta \rceil$ layers of thickness $\delta$:

$$I_j = [\alpha + (j - 1)\delta, \ \alpha + j\delta], \quad j = 1, \ldots, L. \tag{37}$$

Let $\mathcal{N}_j = \{ x \in A_n(x) : \|f(x)\| \in I_j \}$ and the sample set $\mathcal{M}$ can be divided into layers based on the feature norm, where the $j$-th layer is given by:

$$\mathcal{M}_j = \sup\{|S_j| : S_j \subset N_j, \forall x, x^+ \in S, \|f(x) - f(x^+)\| \geq \delta\} \tag{38}$$

Consequently, we have $M = \sum_{j=1}^{L} M_j$ which is the number of points that can be accommodated in the interval $(\alpha, \beta)$ under the separability constraint.

Next, arbitrarily select two points $z_1 = f(x_1)$ and $z_2 = f(x_2)$ from the same layer $I_j$, for $u, v \in S^{n-1}$, $< u, v > = \cos\theta$, $r_1, r_2 \in I_j$ and $|r_1 - r_2| \leq \delta$, we know:

$$\|z_1 - z_2\| = \sqrt{(r_1 - r_2)^2 + 2r_1 r_2 (1 - \cos\theta)} = \sqrt{(r_1 - r_2)^2 + 4r_1 r_2 \sin^2(\frac{\theta}{2})}, \quad \theta \in [0, \pi] \tag{39}$$

It can be observed that the minimum of $\|z_1 - z_2\|$ occurs at $r_1 = r_2$, that is, $z_1$ and $z_2$ are the most difficult to separate when $r_1 = r_2$. At this point, we have $\|z_1 - z_2\| = 2r_1 \sin(\theta/2)$. To ensure that $\|z_1 - z_2\| \geq \delta$, it suffices to satisfy the following condition:

$$2r \sin(\frac{\theta}{2}) \geq \delta \implies \theta \geq 2\arcsin\frac{\delta}{2r} \geq 2\arcsin\frac{\delta}{2r^{\max}} \tag{40}$$

For the $j$-th layer, we have:

$$\theta_j \geq 2\arcsin\frac{\delta}{2r_j^{\max}} \geq 2\arcsin\frac{\delta}{2(\alpha + j\delta)} \geq 2\arcsin\frac{\delta}{2\beta} \tag{41}$$

Thus, for the $j$-th layer with interval $[\alpha + (j-1)\delta, \alpha + j\delta]$, the angle between any two points within this interval satisfies: $\theta_j \geq 2\arcsin\frac{\delta}{2\beta}$

Finally, according to spherical code theory(Ericson & Zinoviev, 2001), $A(n, \theta)$ represents the maximum number of points that can be placed on an $n$-dimensional sphere such that any two points are separated by an angle of at least $\theta$; this is also referred to as the spherical code size(Kabatiansky & Levenshtein, 1978).

From Theorem 4 in Cohn & Zhao (2014), we know that:

$$\frac{1}{n}\log A(n, \theta) \lesssim \frac{1 + \sin\theta}{2\sin\theta}\log\frac{1 + \sin\theta}{2\sin\theta} - \frac{1 - \sin\theta}{2\sin\theta}\log\frac{1 - \sin\theta}{2\sin\theta}. \tag{42}$$

We know $A(n, \theta)$ decreases monotonically with respect to $\theta$ and set $R(\theta) = \frac{1+\sin\theta}{2\sin\theta}\log\frac{1+\sin\theta}{2\sin\theta} - \frac{1-\sin\theta}{2\sin\theta}\log\frac{1-\sin\theta}{2\sin\theta}$.

By definition, we have:

$$\mathcal{M} = \sum_{j=1}^{L}\mathcal{M}_j \leq \sum_{j=1}^{L} A(n, \theta_j) \leq L \cdot \max_{1 \leq j \leq L} A(n, \theta_j) \leq L \cdot A(n, 2\arcsin\frac{\delta}{2\beta}) \leq \left\lceil\frac{\beta - \alpha}{\delta}\right\rceil \cdot \exp(nR(2\arcsin\frac{\delta}{2\beta})) \tag{43}$$

Then, we finish the proof. $\qquad\square$

**Theorem C.3.** *(Recap of theorem 4.6) Let $B_k = \{\exists \ \bar{x}, \bar{x}_1 \in C_k, \|f(\bar{x}) - f(\bar{x}_1)\| < \varepsilon\}$ with $\varepsilon \in (0, \alpha]$ denote the intra-class collapse event, the inter-class connectivity event as $D_k = \{\exists \ \bar{y} \in C_{\neg k}, \bar{x} \in C_k : \bar{y}$ is the augmented-annular neighbor of $\bar{x}\}$ and $W$ denote the class confusion event. When $\mathcal{M} > \left\lceil\frac{\beta-\alpha}{\alpha}\right\rceil \cdot \exp\left(nR(2\arcsin\frac{\alpha}{2\beta})\right)$ and given $P(D_k|B_k) = \rho$, we have:*

$$P(W) \geq \rho \cdot \sum_{k=1}^{K} P(B_k), \tag{44}$$

*where $P(\cdot)$ denotes the probability of the corresponding event.*

*Proof.* When $\delta = \alpha$, according to the Lemma C.2, we have $\mathcal{M} > \left\lceil\frac{\beta-\alpha}{\alpha}\right\rceil \cdot \exp\left(nR(2\arcsin\frac{\alpha}{2\beta})\right)$ and given $P(D_k|B_k) = \rho$. Then for $x \in A(\bar{x})$, there necessarily exists a positive sample set $S \subset A(\bar{x})$ that does not satisfy $\alpha$-$\beta$ connectivity. That is, there exists $x, x^+ \in S$ such that $\|f(x) - f(x^+)\| \leq \alpha$.

Furthermore, according to the Assumption 4.2 and Definition 4.1, there must exist at least the intra-class sample $\bar{x}, \bar{x}_1$ that fails to satisfy the $\alpha$-$\beta$ connectivity, i.e., $\|f(\bar{x}) - f(\bar{x}_1)\| \leq \alpha$. For $\varepsilon \leq \alpha$ and $\|f(\bar{x}) - f(\bar{x}_1)\| \leq \varepsilon$, that is, when the positive sample set $S$ violates the $\alpha$-$\beta$ connectivity connectivity condition, it indicates the occurrence of a intra-class local feature collapse:

$$B_k = \{\exists \ \bar{x}, \bar{x}_1 \in C_k, 0 < \varepsilon \leq \alpha : \|f(\bar{x}) - f(\bar{x}_1)\| < \varepsilon\} \tag{45}$$

Moreover, we know that the global class confusion event $W$ occurs if there is any improper inter-class connectivity. Formally, $W \supseteq \bigcup_{k=1}^{K} D_k$. Since samples from distinct classes are disjoint, we can decompose the probability by summing over the specific collapse-induced confusion events:

$$P(W) \geq P\left(\bigcup_{k=1}^{K}(D_k \cap B_k)\right) = \sum_{k=1}^{K} P(D_k \cap B_k). \tag{46}$$

Applying the definition of conditional probability $P(D_k \cap B_k) = P(D_k \mid B_k)P(B_k)$ and substituting the assumption $P(D_k \mid B_k) = \rho$, we obtain:

$$P(W) \geq \sum_{k}^{K} P(D_k \cap B_k) = \sum_{k}^{K} P(D_k|B_k) \cdot P(B_k) = \rho \cdot \sum_{k}^{K} P(B_k) \tag{47}$$

Then, we finish the proof. $\qquad\square$

### C.3. Proof of Theorem 4.7

**Theorem C.4.** *(Recap of theorem 4.7) For any $f \in F$, its downstream classification risk $\mathcal{L}_{\mathrm{CE}}^{\mu}(f)$ can be bounded by the contrastive learning risk $\mathcal{L}_{\mathrm{NCE}}(f)$ and the connectivity risk $\mathcal{L}_{\eta}(f)$:*

$$\mathcal{L}_{\mathrm{CE}}^{\mu}(f) + \log(m/K) \ \leq \ \mathcal{L}_{\mathrm{NCE}}(f) + b \cdot \sqrt{\mathcal{L}_{\eta}(f)} + \mathcal{O}(\sqrt{N/C} + m^{-1/2}), \tag{48}$$

*where $m$ is the number of negative samples, and $C \geq p-q+1$ (from Theorem 4.4) denotes the number of mutually connected intra-class samples. The coefficient $b = \sqrt{\frac{1}{2(1-\omega)}}$ is determined by the positive pairs correlation $\omega = \mathrm{tr}(\Sigma_{12})/\mathrm{tr}(\Sigma)$.*

To complete the proof of Theorem C.4, we first introduce the following lemmas.

**Lemma C.5.** *For $\bar{x}, \bar{x}_1 \in C_k$ with $|C_k| = N$, and $x \in A(\bar{x})$, $x_1 \in A(\bar{x}_1)$, if $x, x_1 \sim P(\mu, \Sigma)$, we have:*

$$\mathrm{Var}(f(x) \mid y) \leq \frac{1}{2(1-\omega)}(\mathcal{L}_{\eta} + (2N/C + 4)\beta^2), \tag{49}$$

*where $\Sigma_{12} = \mathrm{Cov}(f(x), f(x_1) \mid y)$, $\omega = \mathrm{tr}(\Sigma_{12})/\mathrm{tr}(\Sigma)$ and $\mathrm{Var}(f(x) \mid y) = \mathbb{E}_{p(x,y)}\|f(x) - \mu\|^2$ denotes the conditional feature variance.*

*Proof.* By definition, for $\bar{x}, \bar{x}_1 \in C_k$, we know $x \in A(\bar{x})$, $x_1 \in A(\bar{x}_1)$ are identically distributed but not independent. Therefore, we difine $x, x_1 \sim P(\mu, \Sigma)$. Then,we have:

$$\begin{aligned}
\mathrm{Var}(f(x) \mid y) &= \mathbb{E}_{p(x,y)}\|f(x) - \mu\|^2 \\
&= \mathbb{E}_{p(x,y)}\mathrm{tr}\big[(f(x) - \mu)(f(x) - \mu)^{\top}\big] \\
&= \mathrm{tr}\big[\mathbb{E}_{p(x,y)}(f(x) - \mu)(f(x) - \mu)^{\top}\big] \\
&= \mathrm{tr}(\Sigma),
\end{aligned}$$

and

$$\begin{aligned}
\mathop{\mathbb{E}}_{\substack{\bar{x},\bar{x}_1 \in C_k \\ x \in A(\bar{x}), x_1 \in A(\bar{x}_1)}} \|f(x) - f(x_1)\|^2 &= \mathop{\mathbb{E}}_{\substack{\bar{x},\bar{x}_1 \in C_k \\ x \in A(\bar{x}), x_1 \in A(\bar{x}_1)}} \big[f(x)^{\top}f(x) + f(x_1)^{\top}f(x_1) - 2f(x)^{\top}f(x_1)\big] \\
&= \mathop{\mathbb{E}}_{\substack{\bar{x},\bar{x}_1 \in C_k \\ x \in A(\bar{x}), x_1 \in A(\bar{x}_1)}} \big[(f(x) - \mu)^{\top}(f(x) - \mu) + 2f(x)^{\top}\mu - \|\mu\|^2\big] \\
&\quad + \mathop{\mathbb{E}}_{\substack{\bar{x},\bar{x}_1 \in C_k \\ x \in A(\bar{x}), x_1 \in A(\bar{x}_1)}} \big[(f(x) - \mu)^{\top}(f(x) - \mu) + 2f(x_1)^{\top}\mu - \|\mu\|^2\big] \\
&\quad - 2\mathop{\mathbb{E}}_{\substack{\bar{x},\bar{x}_1 \in C_k \\ x \in A(\bar{x}), x_1 \in A(\bar{x}_1)}} \big[(f(x) - \mu)^{\top}(f(x_1) - \mu) + f(x_1)^{\top}\mu + f(x)^{\top}\mu - \|\mu\|^2\big] \\
&= 2\left(\mathrm{tr}(\Sigma) - \mathrm{tr}(\Sigma_{12})\right).
\end{aligned}$$

Moreover, since $x$ and $x_1$ are positively correlated, we know $0 < \text{tr}(\Sigma_{12}) < \text{tr}(\Sigma)$. Let $\omega = \text{tr}(\Sigma_{12})/\text{tr}(\Sigma)$ denote the degree of correlation between $x$ and $x_1$, we have:

$$\text{Var}(f(x) \mid y) = \frac{1}{2(1-\omega)} \mathbb{E}_{\substack{\bar{x}, \bar{x}_1 \in C_k \\ x \in A(\bar{x}), x_1 \in A(\bar{x}_1)}} \|f(x) - f(x_1)\|^2. \tag{50}$$

Next, let $d(x, x^+) = \|f(x) - f(x_1)\|$, then we further derive the bound:

$$\mathbb{E}_{\substack{\bar{x}, \bar{x}_1 \in C_k \\ x \in A(\bar{x}), x_1 \in A(\bar{x}_1)}} \|f(x) - f(x_1)\|^2 = \mathbb{E}_{\substack{\bar{x}, \bar{x}_1 \in C_k \\ x \in A(\bar{x}), x_1 \in A(\bar{x}_1)}} \|\eta - (\eta - d(x, x_1))\|^2$$

$$\leq \mathbb{E}_{\substack{\bar{x}, \bar{x}_1 \in C_k \\ x \in A(\bar{x}), x_1 \in A(\bar{x}_1)}} \left[ (\eta - d(x, x_1))^2 + 2\eta d(x, x_1) \right]$$

$$\leq \mathcal{L}_\eta + 2\beta \cdot \mathbb{E}_{\substack{\bar{x}, \bar{x}_1 \in C_k \\ x \in A(\bar{x}), x_1 \in A(\bar{x}_1)}} d(x, x_1)$$

Next, by applying the Triangle Inequality and Noting that $\|f(x) - f(\bar{x})\| = \|f(\bar{x}_1) - f(x_1)\| \leq \beta$.

$$\mathbb{E}_{\substack{\bar{x}, \bar{x}_1 \in C_k \\ x \in A(\bar{x}), x_1 \in A(\bar{x}_1)}} d(x, x_1) = \mathbb{E}_{\substack{\bar{x}, \bar{x}_1 \in C_k \\ x \in A(\bar{x}), x_1 \in A(\bar{x}_1)}} \|f(x) - f(\bar{x}) + f(\bar{x}) - f(\bar{x}_1) + f(\bar{x}_1) - f(x_1)\|$$

$$\leq \mathbb{E}_{\substack{\bar{x}, \bar{x}_1 \in C_k \\ x \in A(\bar{x}), x_1 \in A(\bar{x}_1)}} \|f(x) - f(\bar{x})\| + \|f(\bar{x}) - f(\bar{x}_1)\| + \|f(\bar{x}_1) - f(x_1)\|$$

$$\leq 2\beta + \mathbb{E}_{\substack{\bar{x}, \bar{x}_1 \in C_k \\ x \in A(\bar{x}), x_1 \in A(\bar{x}_1)}} \|f(\bar{x}) - f(\bar{x}_1)\|$$

To derive the upper bound on the expected intra-class distance $\mathbb{E}[\|f(\bar{x}) - f(\bar{x}_1)\|]$, we analyze the geometric structure of the class manifold $C_k$ of size $N = |C_k|$ under the constraints of connectivity. We partition the intra-class sample set $C_k \setminus \{\bar{x}\}$ into two disjoint subsets based on their connectivity relative to the anchor $\bar{x}$: a directly connected set $S_c = B_p(x; \alpha, \beta)$ and a remote set $S_r = C_k \setminus (S_c \cup \{\bar{x}\})$.

First, regarding the connected set $S_c$, Theorem 4.4 establishes that its cardinality is lower-bounded by the effective connectivity degree $C = p - q + 1$ (i.e., $|S_c| = C$). For any sample $y \in S_c$, we know:

$$\|f(x) - f(y)\| \leq \beta. \tag{51}$$

Conversely, for the remaining $N-C$ samples in the remote set $S_r$, although a direct connection does not exist, Assumption 4.2 guarantees the existence of a path linking $x$ to any remote sample $z \in S_r$ through the class manifold. Crucially, since every sample in the manifold possesses a local connectivity degree of at least $C$, effectively covering a local "stride" of capacity $\beta$, the manifold can be traversed via a sequence of hops. The maximum number of hops $H$ required to reach any sample $z$ is bounded by the ratio of the total population to the local connectivity coverage, satisfying $H \leq \lceil N/C \rceil < N/C + 1$. Consequently, applying the Triangle Inequality along this path, the distance to any remote sample is bounded by

$$\|f(x) - f(z)\| \leq H \cdot \beta < (\frac{N}{C} + 1)\beta. \tag{52}$$

According to the equation 51 and 52, we can obtain:

$$\mathbb{E}_{\substack{\bar{x}, \bar{x}_1 \in C_k \\ x \in A(\bar{x}), x_1 \in A(\bar{x}_1)}} \|f(\bar{x}) - f(\bar{x}_1)\| \leq P(x_1 \in S_c) \mathbb{E}_{\substack{\bar{x}, \bar{x}_1 \in C_k \\ x \in A(\bar{x}), x_1 \in A(\bar{x}_1)}} \|f(\bar{x}) - f(\bar{x}_1)\|$$

$$+ P(x_1 \in S_r) \mathbb{E}_{\substack{\bar{x}, \bar{x}_1 \in C_k \\ x \in A(\bar{x}), x_1 \in A(\bar{x}_1)}} \|f(\bar{x}) - f(\bar{x}_1)\|$$

$$\leq \frac{C}{N} \cdot \beta + \frac{N - C}{N} \cdot \left( \frac{N}{C} + 1 \right) \beta$$

$$= \frac{N}{C} \beta$$

In conclusion, we obtain the following result:

$$\mathbb{E}_{\substack{\bar{x}, \bar{x}_1 \in C_k \\ x \in A(\bar{x}), x_1 \in A(\bar{x}_1)}} \|f(x) - f(x_1)\|^2 \leq \mathcal{L}_\eta + 2\beta \cdot \mathbb{E}_{\substack{\bar{x}, \bar{x}_1 \in C_k \\ x \in A(\bar{x}), x_1 \in A(\bar{x}_1)}} d(x, x_1)$$

$$\leq \mathcal{L}_\eta + 2\beta \cdot \left( 2\beta + \frac{N}{C} \beta \right)$$

$$\leq \mathcal{L}_\eta + \left( \frac{2N}{C} + 4 \right) \beta^2$$

Combining on Eq. (50), it follows that:

$$\text{Var}(f(x) \mid y) \leq \frac{1}{2(1-\omega)}(\mathcal{L}_\eta + (2N/C + 4)\beta^2) \tag{53}$$

Then, the proof is completed. $\qquad\qquad\square$

The following Lemma C.6 originates from Theorem 4.2 in Wang et al. (2022).

**Lemma C.6.** *If the labels are deterministic (one-hot) and consistent: $p(y|x) = p(y|x^+)$, then for any $f \in \mathcal{F}$, its downstream classification risk $\mathcal{L}_{\text{CE}}^\mu(f)$ can be bounded by the contrastive learning risk $\mathcal{L}_{\text{NCE}}(f)$:*

$$\mathcal{L}_{\text{CE}}^\mu(f) + \log(m/K) \leq \mathcal{L}_{\text{NCE}}(f) + \sqrt{\text{Var}(f(x) \mid y)} + \mathcal{O}(m^{-1/2}),$$

*where $\log(m/K)$ is a constant, and $\mathcal{O}(m^{-1/2})$ represents the order of the approximation error introduced by using $m$ negative samples.*

We now formally prove Theorem C.4.

*Proof of Theorem C.4.* Let $b = \sqrt{\frac{1}{2(1-\omega)}}$ and apply the result of Lemma C.5 to Lemma C.6, we further derive:

$$\mathcal{L}_{\text{CE}}^\mu(f) + \log(m/K) \leq \mathcal{L}_{\text{NCE}}(f) + \sqrt{\text{Var}(f(x) \mid y)} + \mathcal{O}(m^{-1/2})$$

$$\leq \mathcal{L}_{\text{NCE}}(f) + \sqrt{\frac{1}{2(1-\omega)}(\mathcal{L}_\eta + (2N/C + 4)\beta^2)} + \mathcal{O}(m^{-1/2})$$

$$\leq \mathcal{L}_{\text{NCE}}(f) + b \cdot \sqrt{\mathcal{L}_\eta} + b\beta\sqrt{2N/C + 4} + \mathcal{O}(m^{-1/2})$$

$$\leq \mathcal{L}_{\text{NCE}}(f) + b \cdot \sqrt{\mathcal{L}_\eta} + \mathcal{O}(\sqrt{N/C} + m^{-1/2})$$

Then, the proof is completed. $\qquad\qquad\square$

# D. Proof of Section 5

In this section, we provide the proof details of the theorem in the Section 5.

## D.1. Proof of Theorem 5.2

**Theorem D.1.** *(Recap of theorem 5.2) Given an encoder $f$ and a valid smoothing parameter $h \in (0, \min\{\frac{\sqrt{n}L\kappa(2\alpha^3 - 3\beta^2\varepsilon)}{6\beta^2}, \frac{\delta - \beta\max\{4m^2, 2mm', 4m'^2\}}{2\sqrt{n}L\kappa}\})$, the total probability of error caused by overly strong or weak augmentations is bounded by:*

$$R \leq \mathcal{O}(\delta, \varepsilon) \cdot \mathcal{L}_{\alpha-\beta}(f), \tag{54}$$

*where the coefficient $\mathcal{O}(\delta, \epsilon)$ is defined as:*

$$\mathcal{O}(\delta, \varepsilon) = \inf_h \frac{\alpha^2}{\alpha^3 - 3\beta^2(\varepsilon/2 + \sqrt{n}Lh\kappa)} + \frac{\max\{4m^2, 2mm', 4m'^2\}}{\delta - 2\sqrt{n}Lh\kappa - \beta\max\{4m^2, 2mm', 4m'^2\}}. \tag{55}$$

To complete the proof of Theorem D.1, we first introduce the following lemmas.

**Lemma D.2.** *For a given encoder $f$, the alignment $\widehat{\mathcal{L}}_{align}(f)$ admits the following upper and lower bounds, which are controlled by $\widehat{\mathcal{L}}_{\alpha-\beta}(f)$*

$$\alpha - \widehat{\mathcal{L}}_{\alpha-\beta}(f) \leq \widehat{\mathcal{L}}_{align}(f) = \mathbb{E}_{x_1,x_2 \in A(x)}\|f(x_1) - f(x_2)\| \leq \beta + \widehat{\mathcal{L}}_{\alpha-\beta}(f), \tag{56}$$

*where $\widehat{\mathcal{L}}_{\alpha-\beta}(f) = \mathbb{E}_{x_1,x_2 \in A(x)}[(\alpha - \|f(x_1) - f(x_2)\|)_+ + (\|f(x_1) - f(x_2)\| - \beta)_+].$*

*Proof.* Let $d = \|f(x_1) - f(x_2)\|$, for $\widehat{\mathcal{L}}_{\alpha-\beta}(f) + \beta$, we can divide the analysis into the following three cases.

(1) $\alpha < d$, $(\alpha - d)_+ + (d - \beta)_+ + \beta = \alpha - d + \beta \geq d$.

(2) $\alpha \leq d \leq \beta$, $(\alpha - d)_+ + (d - \beta)_+ + \beta = \beta \geq d$.

(3) $d < \beta$, $(\alpha - d)_+ + (d - \beta)_+ + \beta = d - \beta + \beta = d$.

Therefore, we obtain $\mathbb{E}_{x_1, x_2 \in A(x)}[d] \leq \widehat{\mathcal{L}}_{\alpha-\beta}(f) + \beta$.

For $\alpha - \widehat{\mathcal{L}}_{\alpha-\beta}(f)$, we can also divide the analysis into the following three cases.

(1) $\alpha < d$, $\alpha - [(\alpha - d)_+ + (d - \beta)_+] = \alpha - \alpha + d = d$.

(2) $\alpha \leq d \leq \beta$, $\alpha - [(\alpha - d)_+ + (d - \beta)_+] = \alpha \leq d$.

(3) $d < \beta$, $\alpha - [(\alpha - d)_+ + (d - \beta)_+] = \alpha - d + \beta \leq d$.

Therefore, we obtain $\mathbb{E}_{x_1, x_2 \in A(x)}[d] \geq \alpha - \widehat{\mathcal{L}}_{\alpha-\beta}(f)$.

With this, we have completed the proof. $\square$

Next, we proceed based on the Defined measure-theoretic framework and assume that $A_\theta(x)$ varies smoothly with respect to the augmentation parameter $\theta$, satisfying the $L$-Lipschitz condition: $\|A_{\theta_1}(x) - A_{\theta_2}(x)\| \leq L \|\theta_1 - \theta_2\|, \forall x, \theta_1, \theta_2$. Then we have the following result.

Without loss of generality, we assume that both discrete and continuous augmentations are uniformly sampled within their respective spaces.

First, we consider the measure-theoretic representation of uniform sampling for discrete augmentations. For any measurable function $g \colon \mathcal{A}(x) \to \mathbb{R}$ and $x' \in A(x)$, we have

$$
\begin{aligned}
\mathbb{E}_{x' \sim \mu_x^{(d)}}[g(x')] &= \int_{\mathcal{A}_x} g(x') \, d\mu_x^{(d)}(x') \\
&= \frac{1}{m} \sum_{\gamma=1}^{m} \int g(x') \, d\varphi_{A_\gamma(x)}(x') \\
&= \frac{1}{m} \sum_{\gamma=1}^{m} g(A_\gamma(x)).
\end{aligned}
$$

Equivalently, this can be viewed as drawing one augmentation at random from the $m$ discrete candidates with probability $1/m$.

Next, we consider the measure-theoretic representation of uniform sampling for continuous augmentations. For any measurable function $g \colon \mathcal{A}(x) \to \mathbb{R}$ and $x' \in A(x)$, we have

$$
\begin{aligned}
\mathbb{E}_{x' \sim \mu_x^{(c)}}[g(x')] &= \int_{\mathcal{A}_x} g(x') \, d\mu_x^{(c)}(x') \\
&= \int_{\theta \in \Theta} g(A_\theta(x)) \, d\mu_x^{(c)}(A_\theta(x)) \\
&= \int_{\theta \in \Theta} g(A_\theta(x)) p_{A_\theta(x)} \, d\theta.
\end{aligned}
$$

We partition the parameter domain $\Theta$ into $m'$ equal-volume cells $\{\Theta_j\}_{j=1}^{m'}$, each of volume approximately $h^n$, where $m' = \text{vol}(\Theta)/h^n$. For any $x$, the density $p_{A_\theta(x)}$ is approximately uniform, we set $p_{A_\theta(x)} = 1/\text{vol}(\Theta)$. Then, we have

$$
\begin{aligned}
\int_{\theta \in \Theta} g(A_\theta(x)) p_{A_\theta(x)} \, d\theta &= \sum_{j=1}^{m'} \int_{\theta \in \Theta_j} g(A_\theta(x)) p_{A_\theta(x)} \, d\theta \\
&= \frac{1}{m'} \sum_{j=1}^{m'} \int_{\theta \in \Theta_j} \frac{1}{h^n} g(A_\theta(x)) \, d\theta.
\end{aligned}
$$

We can now decompose the alignment term. For any given $x_1, x_2 \in A(x)$ and measurable bivariate function $G(x_1, x_2) = \|f(x_1) - f(x_2)\|$, we have

$$
\begin{aligned}
\mathbb{E}_{x_1,x_2 \in A(x)}[G(x_1,x_2)] &= \lambda_d^2 \mathbb{E}_{x_1,x_2 \sim \mu_x^{(d)}}[G(x_1,x_2)] + 2\lambda_d\lambda_c \mathbb{E}_{x_1 \sim \mu_x^{(d)}, x_2 \sim \mu_x^{(c)}}[G(x_1,x_2)] + \lambda_c^2 \mathbb{E}_{x_1,x_2 \sim \mu_x^{(c)}}[G(x_1,x_2)] \\
&= \lambda_d^2 \int_{\mathcal{A}_x} \int_{\mathcal{A}_x} G(x_1,x_2)\, d\mu_x^{(d)}(x_1)\, d\mu_x^{(d)}(x_2) + 2\lambda_d\lambda_c \int_{\mathcal{A}_x} \int_{\mathcal{A}_x} G(x_1,x_2)\, d\mu_x^{(d)}(x_1)\, d\mu_x^{(c)}(x_2) \\
&\quad + \lambda_c^2 \int_{\mathcal{A}_x} \int_{\mathcal{A}_x} G(x_1,x_2)\, d\mu_x^{(c)}(x_1)\, d\mu_x^{(c)}(x_2) \\
&= \underbrace{\lambda_d^2 \frac{1}{m^2} \sum_{\gamma=1}^{m} \sum_{\beta=1}^{m} G(A_\gamma(x), A_\beta(x))}_{\Lambda_1} + \underbrace{2\lambda_d\lambda_c \frac{1}{mm'} \sum_{\gamma=1}^{m} \sum_{j=1}^{m'} \int_{\theta \in \Theta_j} \frac{1}{h^{2n}} G(A_\gamma(x), A_\theta(x))}_{\Lambda_2} \\
&\quad + \underbrace{\lambda_c^2 \frac{1}{m'^2} \sum_{i=1}^{m'} \sum_{j=1}^{m'} \int_{\theta_1 \in \Theta_i} \int_{\theta_2 \in \Theta_j} \frac{1}{h^n} G(A_{\theta_1}(x), A_{\theta_2}(x))}_{\Lambda_3}.
\end{aligned}
$$

In the following, we assume that the probabilities of selecting discrete and continuous augmentations are the same, i.e., $\lambda_d = \lambda_c = 1/2$. We then have the following Lemma D.3 and Lemma D.4.

**Lemma D.3.** *For a given encoder $f$ and $h \in \left(0, \frac{\delta - \beta \max\{4m^2, 2mm', 4m'^2\}}{2\sqrt{n}L h\kappa}\right)$, we have*

$$R_\delta \leq \Phi(\delta) \cdot \mathcal{L}_{\alpha-\beta}(f), \tag{57}$$

*where $\Phi(\delta) = \frac{\max\{4m^2, 2mm', 4m'^2\}}{\delta - 2\sqrt{n}L h\kappa - \beta \max\{4m^2, 2mm', 4m'^2\}}$.*

*Proof.* The following result can be derived from the proof of Theorem 2 in Huang et al. (2021) and Lemma D.2:

$$
\begin{aligned}
\sup_{x_1,x_2 \in A(x)} \|f(x_1) - f(x_2)\| &\leq \max\{4m^2, 2mm', 4m'^2\} \mathbb{E}_{x_1,x_2 \in A(x)} \|f(x_1) - f(x_2)\| + 2\sqrt{n}\,L h\kappa \\
&\leq \max\{4m^2, 2mm', 4m'^2\} (\widehat{\mathcal{L}}_{\alpha-\beta}(f) + \beta) + 2\sqrt{n}\,L h\kappa
\end{aligned}
$$

Thus, the following set $S$ is a subset of $S_\delta = \{x : \sup_{x_1,x_2 \in A(x)} \|f(x_1) - f(x_2)\| \leq \delta\}$:

$$S = \left\{ x : \widehat{\mathcal{L}}_{\alpha-\beta}(f) \leq \frac{\delta - 2\sqrt{n}\,L h\kappa}{\max\{4m^2, 2mm', 4m'^2\}} - \beta \right\} \subseteq S_\delta. \tag{58}$$

Then by Markov's inequality, we have

$$
\begin{aligned}
R_\delta = \mathbb{P}[\overline{S_\delta}] &\leq \mathbb{P}[\overline{S}] \\
&\leq \frac{\mathbb{E}_{x \in C_k}(\widehat{\mathcal{L}}_{\alpha-\beta}(f))}{\frac{\delta - 2\sqrt{n}L h\kappa}{\max\{4m^2, 2mm', 4m'^2\}} - \beta} \\
&= \frac{\max\{4m^2, 2mm', 4m'^2\}}{\delta - 2\sqrt{n}L h\kappa - \beta \max\{4m^2, 2mm', 4m'^2\}} \mathcal{L}_{\alpha-\beta}(f)
\end{aligned}
$$

Thus, for all $h \in \left(0, \frac{\delta - \beta \max\{4m^2, 2mm', 4m'^2\}}{2\sqrt{n}L\kappa}\right)$, we have

$$
\begin{aligned}
R_\delta &\leq \inf_h \frac{\max\{4m^2, 2mm', 4m'^2\}}{\delta - 2\sqrt{n}L h\kappa - \beta \max\{4m^2, 2mm', 4m'^2\}} \mathcal{L}_{\alpha-\beta}(f) \\
&= \Phi(\delta) \cdot \mathcal{L}_{\alpha-\beta}(f).
\end{aligned}
$$

This finishes the proof. $\qquad \square$

**Lemma D.4.** *For a given encoder $f$ and $h \in \left(0, \frac{\sqrt{n}L\kappa(2\alpha^3 - 3\beta^2\varepsilon)}{6\beta^2}\right)$, we have*

$$R_\varepsilon \leq \Psi(\varepsilon) \cdot \mathcal{L}_{\alpha-\beta}(f), \tag{59}$$

*where $\Psi(\varepsilon) = \inf_h \frac{\alpha^2}{\alpha^3 - 3\beta^2(\varepsilon/2 + \sqrt{n}Lh\kappa)}$.*

*Proof.* For any given $\theta$,

$$\inf_{\theta'} \|f(A_\gamma(x)) - f(A_{\theta'}(x))\| \geq \|f(A_\gamma(x)) - f(A_\theta(x))\| - \sup_{\theta'} \|f(A_\theta(x)) - f(A_{\theta'}(x))\|$$

$$\geq \|f(A_\gamma(x)) - f(A_\theta(x))\| - \sup_{\theta_1, \theta_2} \|f(A_{\theta_1}(x)) - f(A_{\theta_2}(x))\|.$$

Then, for $\gamma \in [m], j \in [m']$, we have

$$\inf_{\theta' \in \Theta_j} \|f(A_\gamma(x)) - f(A_{\theta'}(x))\|$$

$$= \int_{\Theta_j} \frac{1}{h^n} \inf_{\theta' \in \Theta_j} \|f(A_\gamma(x)) - f(A_{\theta'}(x))\| \, d\theta$$

$$\geq \int_{\Theta_j} \frac{1}{h^n} \|f(A_\gamma(x)) - f(A_\theta(x))\| \, d\theta - \sup_{\theta_1, \theta_2 \in \Theta_j} \|f(A_{\theta_1}(x)) - f(A_{\theta_2}(x))\|$$

$$\geq \int_{\Theta_j} \frac{1}{h^n} \|f(A_\gamma(x)) - f(A_\theta(x))\| \, d\theta - \kappa \sup_{\theta_1, \theta_2 \in \Theta_j} \|A_{\theta_1}(x) - A_{\theta_2}(x)\|$$

$$\geq \int_{\Theta_j} \frac{1}{h^n} \|f(A_\gamma(x)) - f(A_\theta(x))\| \, d\theta - L\kappa \sup_{\theta_1, \theta_2 \in \Theta_j} \|\theta_1 - \theta_2\|$$

$$= \int_{\Theta_j} \frac{1}{h^n} \|f(A_\gamma(x)) - f(A_\theta(x))\| \, d\theta - L\kappa\sqrt{n}h$$

$$= \int_{\Theta_j} \frac{1}{h^n} \|f(A_\gamma(x)) - f(A_\theta(x))\| \, d\theta - \sqrt{n}Lh\kappa.$$

In a similar manner, we can derive the following result

$$\inf_{\theta \in \Theta_i, \theta' \in \Theta_j} \|f(A_\theta(x)) - f(A_{\theta'}(x))\|$$

$$= \int_{\Theta_i} \int_{\Theta_j} \frac{1}{h^{2n}} \inf_{\theta \in \Theta_i, \theta' \in \Theta_j} \|f(A_\theta(x)) - f(A_{\theta'}(x))\| \, d\theta_2 \, d\theta_1$$

$$\geq \int_{\Theta_i} \int_{\Theta_j} \frac{1}{h^{2n}} \|f(A_{\theta_1}(x)) - f(A_{\theta_2}(x))\| \, d\theta_2 \, d\theta_1 - \int_{\Theta_i} \int_{\Theta_j} \frac{1}{h^{2n}} \sup_{\theta \in \Theta_i} \|f(A_\theta(x)) - f(A_{\theta_1}(x))\| \, d\theta_2 \, d\theta_1$$

$$- \int_{\Theta_i} \int_{\Theta_j} \frac{1}{h^{2n}} \sup_{\theta' \in \Theta_j} \|f(A_{\theta_2}(x)) - f(A_{\theta'}(x))\| \, d\theta_2 \, d\theta_1$$

$$\geq \int_{\Theta_i} \int_{\Theta_j} \frac{1}{h^{2n}} \|f(A_{\theta_1}(x)) - f(A_{\theta_2}(x))\| \, d\theta_2 \, d\theta_1$$

$$- \sup_{\theta, \theta' \in \Theta_i} \|f(A_\theta(x)) - f(A_{\theta'}(x))\| - \sup_{\theta, \theta' \in \Theta_j} \|f(A_\theta(x)) - f(A_{\theta'}(x))\|$$

$$\geq \int_{\Theta_i} \int_{\Theta_j} \frac{1}{h^{2n}} \|f(A_{\theta_1}(x)) - f(A_{\theta_2}(x))\| \, d\theta_2 \, d\theta_1 - 2\sqrt{n}Lh\kappa.$$

Therefore, we can obtain

$$\inf_{x_1,x_2 \in A(x)} \|f(x_1) - f(x_2)\|$$

$$= \min \left\{ \begin{array}{l} \inf_{\gamma,\beta \in [m]} \|f(A_\gamma(x)) - f(A_\beta(x))\| \\ \inf_{\gamma \in [m],\, j \in [m']} \inf_{\theta' \in \Theta_j} \|f(A_\gamma(x)) - f(A_{\theta'}(x))\| \\ \inf_{i,j \in [m']} \inf_{\theta \in \Theta_i,\, \theta' \in \Theta_j} \|f(A_\theta(x)) - f(A_{\theta'}(x))\| \end{array} \right\}$$

$$\geq \min \left\{ \begin{array}{l} \inf_{\gamma,\beta \in [m]} \|f(A_\gamma(x)) - f(A_\beta(x))\| \\ \inf_{\gamma \in [m],\, j \in [m']} \int_{\Theta_j} \frac{1}{h^n} \|f(A_\gamma(x)) - f(A_\theta(x))\| \, d\theta - \sqrt{n} Lh\kappa \\ \inf_{i,j \in [m']} \int_{\Theta_i} \int_{\Theta_j} \frac{1}{h^{2n}} \|f(A_{\theta_1}(x)) - f(A_{\theta_2}(x))\| \, d\theta_2 \, d\theta_1 - 2\sqrt{n} Lh\kappa \end{array} \right\}$$

$$\geq \min \left\{ \begin{array}{l} 4\Delta_1 \cdot \Lambda_1 \\ 2\Delta_2 \cdot \Lambda_2 \\ 4\Delta_3 \cdot \Lambda_3 \end{array} \right\} - 2\sqrt{n} Lh\kappa$$

$$\geq 2\frac{\alpha}{\beta} \cdot \min \left\{ \begin{array}{l} \Lambda_1 \\ \Lambda_2 \\ \Lambda_3 \end{array} \right\} - 2\sqrt{n} Lh\kappa$$

$$\geq 2\frac{\alpha}{\beta} \cdot \frac{\alpha}{\beta} \cdot \frac{(\Lambda_1 + \Lambda_2 + \Lambda_3)}{3} - 2\sqrt{n} Lh\kappa$$

$$= \frac{2\alpha^2}{3\beta^2} \cdot \mathbb{E}_{x_1,x_2 \in A(x)} \|f(x_1) - f(x_2)\| - 2\sqrt{n} Lh\kappa,$$

where $\min\{\Lambda_1, \Lambda_2, \Lambda_3\} \geq \frac{\alpha}{\beta} \cdot \max\{\Lambda_1, \Lambda_2, \Lambda_3\} \geq \frac{\alpha}{\beta} \cdot \frac{(\Lambda_1 + \Lambda_2 + \Lambda_3)}{3}$, $\Delta_1 = \frac{\inf \|f(A_\gamma(x)) - f(A_\beta(x))\|}{\sup \|f(A_\gamma(x)) - f(A_\beta(x))\|} \geq \frac{\alpha}{\beta}$, $\Delta_2 = \frac{\inf \int_{\Theta_j} \frac{1}{h^n} \|f(A_\gamma(x)) - f(A_\theta(x))\|}{\sup \int_{\Theta_j} \frac{1}{h^n} \|f(A_\gamma(x)) - f(A_\theta(x))\|} \geq \frac{\alpha}{\beta}$, and $\Delta_3 = \frac{\inf \int_{\Theta_i} \int_{\Theta_j} \frac{1}{h^{2n}} \|f(A_{\theta_1}(x)) - f(A_{\theta_2}(x))\| \, d\theta_2 \, d\theta_1}{\sup \int_{\Theta_i} \int_{\Theta_j} \frac{1}{h^{2n}} \|f(A_{\theta_1}(x)) - f(A_{\theta_2}(x))\| \, d\theta_2 \, d\theta_1} \geq \frac{\alpha}{\beta}$.

The following result can be derived from the Lemma D.2:

$$\inf_{x_1,x_2 \in A(x)} \|f(x_1) - f(x_2)\| \geq \frac{2\alpha^2}{3\beta^2} \cdot \mathbb{E}_{x_1,x_2 \in A(x)} \|f(x_1) - f(x_2)\| - 2\sqrt{n} Lh\kappa$$

$$\geq \frac{2\alpha^2}{3\beta^2} (\alpha - \widehat{\mathcal{L}}_{\alpha-\beta}(f)) - 2\sqrt{n} Lh\kappa$$

Thus, the following set $S$ is a subset of $S_\varepsilon = \{x : \sup_{x_1,x_2 \in A(x)} \|f(x_1) - f(x_2)\| \geq \varepsilon\}$:

$$S = \left\{ x : \widehat{\mathcal{L}}_{\alpha-\beta}(f) \leq \frac{\alpha^3 - 3\beta^2(\varepsilon/2 + \sqrt{n} Lh\kappa)}{\alpha^2} \right\} \subseteq S_\varepsilon. \tag{60}$$

Then by Markov's inequality, we have

$$R_\varepsilon = \mathbb{P}[\overline{S_\varepsilon}] \leq \mathbb{P}[\overline{S}]$$

$$\leq \frac{\mathbb{E}_{x \in C_k}(\widehat{\mathcal{L}}_{\alpha-\beta}(f))}{\frac{\alpha^3 - 3\beta^2(\varepsilon/2 + \sqrt{n} Lh\kappa)}{\alpha^2}}$$

$$= \frac{\alpha^2}{\alpha^3 - 3\beta^2(\varepsilon/2 + \sqrt{n} Lh\kappa)} \mathcal{L}_{\alpha-\beta}(f).$$

Thus, for all $h \in \left(0, \frac{\sqrt{n}L\kappa(2\alpha^3 - 3\beta^2\varepsilon)}{6\beta^2}\right)$, we have

$$R_\varepsilon \leq \inf_h \frac{\alpha^2}{\alpha^3 - 3\beta^2(\varepsilon/2 + \sqrt{n}Lh\kappa)}\mathcal{L}_{\alpha-\beta}(f)$$
$$= \Psi(\varepsilon) \cdot \mathcal{L}_{\alpha-\beta}(f).$$

This finishes the proof.

We now prove Theorem D.2.

$$R = R_\delta \cup R_\varepsilon \leq R_\delta + R_\varepsilon \leq [\Phi(\delta) + \Psi(\varepsilon)] \cdot \mathcal{L}_{\alpha-\beta}(f) \leq \mathcal{O}(\delta, \varepsilon) \cdot \mathcal{L}_{\alpha-\beta}(f) \tag{61}$$

Then, we finishes the proof. □

## E. Experimental Details and Supplementary Analysis

In this section, we provide detailed ablation studies and supplementary experimental results to further analyze the properties of our proposed framework.

### E.1. Sensitivity to the Regularization Coefficient $\lambda$

The coefficient $\lambda$ in Eq. (5) serves as a critical hyper-parameter, controlling the trade-off between the semantic discrimination of the InfoNCE objective and the structural constraints imposed by our topology-aware connectivity loss.

**Experimental Setup.** To evaluate the impact of this regularization strength, we trained ResNet-50 models on both CIFAR-10 and CIFAR-100 datasets for 100 epochs, varying $\lambda \in \{0, 0.001, 0.01, 0.1\}$. Crucially, for these experiments, we fixed the number of positive samples at $n = 4$ and the target connectivity radius at $\eta = 0.5$. The detailed justifications and ablation studies supporting the selection of these structural hyperparameters are provided in Section E.2 and Section E.3, respectively. Note that $\lambda = 0$ corresponds to the standard SimCLR baseline.

*Table 4.* Linear evaluation accuracy (%) on CIFAR-10 and CIFAR-100 with different regularization coefficients $\lambda$.

| $\lambda$ | 0 | 0.001 | 0.01 | 0.1 |
|---|---|---|---|---|
| CIFAR-10 | 84.33 | 85.06 | **85.47** | 82.11 |
| CIFAR-100 | 60.38 | 60.59 | **62.47** | 56.84 |

**Results and Analysis.** Table 4 presents the linear evaluation accuracy under different coefficients. We observe a consistent trend across both datasets: The downstream performance increases steadily as $\lambda$ increases from 0 to 0.01. Specifically, at $\lambda = 0.01$, the model achieves its peak performance, outperforming the baseline ($\lambda = 0$) by a significant margin. This confirms that introducing explicit topological connectivity effectively regulates the latent space, preventing the topology-agnostic confusion discussed in Section 3. However, setting an overly large coefficient ($\lambda = 0.1$) hurts performance. We attribute this to the fact that excessive regularization may overshadow the primary contrastive task. When the topology loss dominates the optimization landscape, the encoder prioritizes satisfying geometric constraints over learning the fine-grained instance discrimination required for classification, leading to a suboptimal representation.

Based on these results, we adopt $\lambda = 0.01$ as the default setting for all main experiments.

### E.2. Impact of Positive Sample Size $n$ and k-NN Purity Definition

**Selection of Positive Sample Size.** The number of augmented views $n$ plays a pivotal role in defining the topological density of the latent space. While standard contrastive learning typically operates on pairs ($n = 2$), our framework requires a higher connectivity degree to approximate local manifold structures.

For a given input sample $x_i$, we generate $n$ augmented views $\mathcal{B}_i = \{x_0, x_1, \ldots, x_{n-1}\}$. We designate $x_0$ as the anchor. To enforce topological constraints. To strictly enforce connectivity constraints across these $n$ views, we extend the formulation

provided in Eq. (8). The generalized total objective $\mathcal{L}_{\eta-\text{NCE}}(f)$ aggregates the loss over all anchor-positive pairs:

$$
\mathbb{E}_{\substack{\bar{x}\sim\hat{\mathcal{P}}_X \\ \{x_0,\ldots,x_{n-1}\}\sim\mathcal{A}(\bar{x})}} \mathbb{E}_{\substack{\{\bar{x}_i^-\}_{i=1}^m\sim\hat{\mathcal{P}}_X \\ x_{1:m}^-\sim\mathcal{A}(\bar{x}_{1:m}^-)}} \left[ \frac{1}{n-1} \sum_{k=1}^{n-1} \left( -\log \frac{\exp(f(x_0)^\top f(x_k))}{\sum_{j=1}^m \exp(f(x_0)^\top f(x_j^-))} + \lambda\left(\eta - \|f(x_0) - f(x_k)\|\right)^2 \right) \right]
\tag{62}
$$

Here, the outer summation averages the loss over all $n-1$ positive pairs. For each pair $(x_0, x_k)$, the first term minimizes the standard InfoNCE loss to ensure semantic alignment, while the second term strictly regulates the pairwise distance to maintain the target topological connectivity $\eta$.

We empirically selected $n=4$ as the optimal configuration. Increasing $n$ beyond 4 yields diminishing returns while quadratically increasing the computational overhead. Conversely, using $n=2$ reduces the topology to a single edge, which is insufficient to form the higher-order connected components required to robustly bridge intra-class variations. Consequently, we adopt $n=4$ as the default setting for all experiments reported in this work.

**Definition of k-NN Purity.** To quantitatively assess the local topological consistency of the learned representations, we employ the k-NN Purity metric. For a dataset of size $n$, let $z_i = f(x_i)$ denote the feature vector of sample $i$, and $\mathcal{N}_k(z_i)$ denote the set of indices of its $k$ nearest neighbors in the latent space. The k-NN Purity is defined as the average proportion of neighbors that share the same class label as the anchor:

$$
\text{Purity}(k) = \frac{1}{n} \sum_{i=1}^n \left( \frac{1}{k} \sum_{j\in\mathcal{N}_k(z_i)} \mathbb{I}(y_i = y_j) \right)
$$

where $y_i$ is the ground-truth label of sample $i$, and $\mathbb{I}(\cdot)$ is the indicator function. A higher purity score indicates that the local manifold structure is semantically consistent, with samples of the same class clustering tightly together; crucially, this also implies that the inter-class boundaries are well-preserved, thereby effectively mitigating the class confusion phenomenon.

### E.3. Impact of Branch Dimension $B$ and Radius $\eta$

**Sensitivity and Selection Strategy for Radius $\eta$.** To systematically evaluate the impact of the connectivity radius $\eta$, we first isolate its effect over a wide spectrum under a fixed baseline configuration (setting the branch dimension $B = 128$). The empirical results for k-NN purity and linear evaluation accuracy are summarized in Table 5.

*Table 5.* Comparison of k-NN purity and linear evaluation accuracy (%) on CIFAR-100 under varying connectivity radius $\eta$ (with $B = 128$).

| $\eta$ | 0 | 0.01 | 0.1 | 0.5 | 1.0 | 1.5 | 2.0 |
|---|---|---|---|---|---|---|---|
| k-NN Purity | 57.70 | 57.95 | 59.10 | 58.83 | 50.35 | 41.55 | 31.44 |
| Accuracy | 58.12 | 58.33 | 60.94 | 60.95 | 57.62 | 50.86 | 47.32 |

As indicated by the empirical trajectory in Table 5, the optimal selection for $\eta$ falls squarely between 0.1 and 0.5, where both k-NN purity and downstream linear accuracy achieve their peak performance. This behavior perfectly mirrors our theoretical framework, which dictates that maintaining representation connectivity within a well-bounded interval prevents premature manifold entanglement. When $\eta$ is close to 0, the structural constraint becomes negligible, causing the framework to degenerate toward standard topology-agnostic contraction. Conversely, an excessively large radius ($\eta \geq 1.0$) severely disrupts the latent space, triggering a drastic drop in k-NN purity down to 31.44% at $\eta = 2.0$. This collapse occurs because an over-extended radius forces distinct semantic clusters to inadvertently bridge and merge into a single connected component.

**Joint Impact of Branch Dimension $B$ and Radius $\eta$.** Having established the optimal range for $\eta$, we further investigate its joint effect with the branch dimension. To effectively capture local manifold structures, we decompose the total projection dimension $D$ into independent subspaces of dimension $B$ (Hofer et al., 2019). The topological regularization is enforced within each subspace and aggregated across the $D/B$ branches. The generalized objective is defined as:

$$
\mathcal{L}_{\eta\text{-NCE}}(f) = \mathcal{L}_{\text{NCE}}(f) + \lambda \sum_{j=1}^{D/B} \left[ \frac{1}{n-1} \sum_{k=1}^{n-1} \left( \eta - \left\| z_0^{(j)} - z_k^{(j)} \right\| \right)^2 \right]
\tag{63}
$$

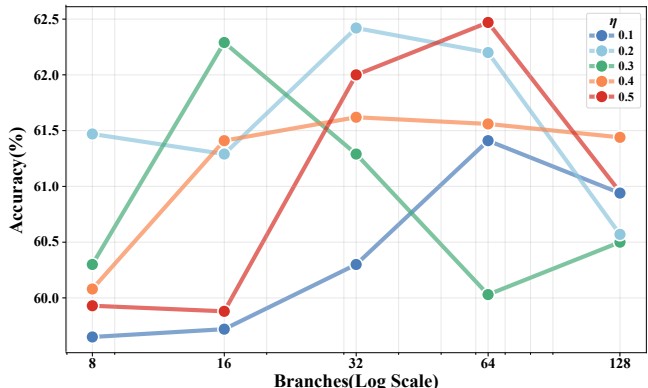

*Figure 3.* Joint sensitivity analysis of Branch Dimension $B$ and Connectivity Radius $\eta$ on CIFAR-100.

We perform a grid search on CIFAR-100 with $\eta \in \{0.1, 0.2, 0.3, 0.4, 0.5\}$ and $B \in \{8, 16, 32, 64, 128\}$. As illustrated in Figure 3, performance exhibits a distinct inverted U-shaped curve, consistently peaking at intermediate dimensions ($B \in \{16, 32, 64\}$). This indicates that moderate granularity offers an optimal trade-off between local structural sensitivity and semantic retention. In contrast, performance degrades at the extremes: at $B = 8$, subspaces lack sufficient capacity to encode meaningful features, leading to fragmented constraints; conversely, at $B = 128$, the metric suffers from the curse of dimensionality and loses the stabilizing "ensemble effect" provided by multiple independent branches.

### E.4. Architectural Generalization and Comparisons with Alternative Methods

**Architectural Generalization.** To demonstrate that our Topology-Aware Contrastive Learning framework is architecture-independent, we conduct additional pre-training experiments using a lighter ResNet-18 backbone on CIFAR-100. The quantitative results are summarized in Table 6.

*Table 6.* Linear evaluation accuracy and k-NN purity (%) comparison using a ResNet-18 backbone on CIFAR-100.

| Backbone | Method | k-NN Purity | Accuracy |
|---|---|---|---|
| ResNet-18 | SimCLR | 38.37 | 52.02 |
| | **Ours** | **41.07** | **55.06** |

**Comparison with Data-Centric Alternative Methods.** Existing paradigms, such as Debiased Contrastive Learning (DCL) (Chuang et al., 2020) or ArCL (Zhao et al., 2023), attempt to address false positive pairs or semantic drift primarily at the data distribution level. They rely heavily on instance selection, filtering, or heuristic re-weighting. In contrast, our framework does not alter or prune the input data streams. Instead, it operates fundamentally at the latent topological level, dynamically restructuring the spatial mechanism by which positive pairs are bound. This provides a principled geometric advantage over purely data-centric operations.

To substantiate this advantage, we evaluate standard SimCLR, DCL, ArCL, and our method on CIFAR-100. We examine their intrinsic latent structure (via k-NN purity) and their robustness against extreme augmentation regimes. Specifically, we report the downstream linear classification performance under the Weak (W) and Strong (S) photometric augmentation configurations defined in Section 6.2. The results are presented in Table 7.

*Table 7.* Performance comparison with alternative methods on CIFAR-100 under extreme photometric augmentations. "S" and "W" denote the Strong and Weak augmentation regimes, respectively.

| Method | k-NN Purity (%) | S-Top1 (%) | S-Top5 (%) | W-Top1 (%) | W-Top5 (%) |
|---|---|---|---|---|---|
| SimCLR | 57.73 | 11.33 | 31.45 | 9.10 | 25.03 |
| DCL (Chuang et al., 2020) | 57.77 | 12.61 | 34.22 | 6.37 | 20.06 |
| ArCL (Zhao et al., 2023) | 59.77 | 12.51 | 32.90 | 10.08 | 27.04 |
| **Ours** | **60.36** | **17.35** | **39.49** | **12.56** | **32.18** |

