# OpenReview forum: "Topology-Aware Contrastive Learning: Regulating Representation Connectivity via Persistent Homology"
_ICML.cc/2026/Conference — ICML 2026 regular_

### Official Review · Reviewer_axD4 · 2026-03-10

**Soundness:** 2
**Presentation:** 3
**Significance:** 2
**Originality:** 2
**Overall Recommendation:** 4
**Confidence:** 3

**Summary:**

This paper proposes a theoretical framework to analyze the topology-agnostic confusion phenomenon in contrastive learning. The authors discuss the trade-off in augmentation strength: overly aggressive augmentations may lead to inter-class connectivity, while overly weak augmentations may break intra-class connectivity. To address this issue, the paper introduces a penalty term that explicitly controls connectivity strength. Empirical results demonstrate that the proposed method improves performance across different augmentation settings.

**Compliance With Llm Reviewing Policy:**

Affirmed.

**Final Justification:**

My main concern was the originality of the paper. The proposed connectivity risk appeared similar to the concept of labeling error in previous work, which initially lowered my assessment of the paper’s contribution. During the rebuttal phase, however, the authors clearly distinguished their concept from prior proposals. This addressed my main concerns, and I therefore raised my score to 4.

**Key Questions For Authors:**

Please see Weakness.

**Limitations:**

Please see Weakness.

**Strengths And Weaknesses:**

**Strengths**

1. The selection of augmentation strength is a central question in contrastive learning. This paper provides a new perspective by explicitly controlling intra-class and inter-class connectivity through a penalty term.
2. The theoretical analysis appears solid, and the explanations are generally clear and well motivated.
3. The paper is well written and easy to follow.

---

**Weaknesses**

1. It is unclear how the proposed connectivity risk differs from the concept of labeling error studied in prior theoretical work on contrastive learning. In both [1] and [2], the authors derive theoretical guarantees that involve a key term measuring the probability that a positive pair comes from different classes. This term seems closely related to the connectivity risk introduced in this paper. It would be helpful for the authors to theoretically compare these quantities or clarify their relationship.
2. Building on the previous point, the main contribution of this paper appears to lie in how the proposed loss mitigates this risk term. However, a number of existing methods attempt to address similar issues, such as detecting false positive pairs or designing better augmentation strategies. These approaches have already demonstrated empirical improvements. Therefore, it would strengthen the paper to include empirical comparisons with these methods, or to investigate whether they can be combined with the proposed loss for further gains.
3. The experimental evaluation should be conducted on ImageNet-scale datasets to better demonstrate the effectiveness of the method in realistic large-scale settings.
4. The design of the hyperparameter η is not fully discussed. The paper estimates an appropriate connectivity level empirically, but in practice the connectivity structure may vary across different connected components of the graph. Therefore, it would be valuable to explore more principled or adaptive strategies for setting this hyperparameter.

[1] HaoChen, Jeff Z., et al. "Provable guarantees for self-supervised deep learning with spectral contrastive loss." *Advances in neural information processing systems* 34 (2021): 5000-5011.

[2] Zhang, Qi, Yifei Wang, and Yisen Wang. "An Augmentation Overlap Theory of Contrastive Learning." *Journal of Machine Learning Research* 26.228 (2025): 1-42.

---

> ### Author Rebuttal · Authors · 2026-03-31
>
> Dear Reviewer axD4,
>
> We are very grateful for your valuable comments. Our responses are as follows:
>
> **Response to W1:**
>
> While "labeling error" and our "connectivity risk" are closely related, they represent fundamentally different mechanisms. Prior theoretical works treat labeling error as an inherent, static property of the data augmentation distribution in the input space $\mathcal{X}$—an unavoidable noise by which the model is passively affected. Conversely, our connectivity risk $\mathcal{L}_\eta(f)$ acts as an active structural constraint within the learned latent space $\mathbb{R}^d$, directly penalizing pairwise distance deviations from a target radius $\eta$.
>
> Furthermore, Section 5.1 formalizes this "labeling error" concept as the strong augmentation error $R_\delta$. Rather than treating it as an inevitable degradation factor in generalization bounds, Theorem 5.2 proves that $R \le \mathcal{O}(\delta,\epsilon) \cdot \mathcal{L}_{\alpha-\beta}(f)$. This demonstrates that explicitly minimizing the topological connectivity risk strictly bounds the total probability of these augmentation-induced errors.
>
> **Response to W2:**
>
> Existing methods such as detecting false positive pairs [1] or designing better augmentation strategies [2] address similar issues at the data level via instance selection, filtering, or re-weighting. Our method, however, does not attempt to filter the data. Instead, it operates at the latent topology level, fundamentally altering the spatial mechanism by which positive pairs are processed within the representation space. This provides an inherent advantage over purely data-centric operations.
>
> To substantiate this claim, we conducted additional experiments on CIFAR-100 to compare standard SimCLR, ArCL[1]，DCL[2], and our proposed method. We evaluated their impact on the latent space structure (measured via k-NN purity) and their sensitivity to extreme augmentation strategies (Strong vs. Weak augmentations, as defined in Section 6.2; here, we only present the accuracy under weak and strong photometric augmentations).
>
> |  Method  | k-NN Purity (%) |  S-Top1   |  S-Top5   |  W-Top1   |  W-Top5   |
> | :------: | :-------------: | :-------: | :-------: | :-------: | :-------: |
> |  SimCLR  |      57.73      |   11.33   |   31.45   |   9.10    |   25.03   |
> |   DCL    |      57.77      |   12.61   |   34.22   |   6.37    |   20.06   |
> |   ArCL   |      59.77      |   12.51   |   32.90   |   10.08   |   27.04   |
> | **Ours** |    **60.36**    | **17.35** | **39.49** | **12.56** | **32.18** |
>
>
> **Response to W3:**
>
> To better demonstrate the effectiveness of our method in realistic large-scale settings, we conducted experiments on the ImageNet-100 and STL-10 datasets. The linear evaluation accuracy and k-NN Purity（$k=3$） are shown in the table below:
>
> |   Dataset    | k-NN (SimCLR) | k-NN (Ours) | Acc (SimCLR) | Acc (Ours) |
> | :----------: | :-----------: | :---------: | :----------: | :--------: |
> | ImageNet-100 |     30.59     |  **37.19**  |    60.98     | **64.14**  |
> |    STL-10    |     74.59     |  **75.27**  |    82.47     | **83.21**  |
>
> ImageNet is characterized by a vast number of fine-grained categories, which inherently entails a high risk of semantic overlap and manifold entanglement. Consequently, our theoretical analysis (Theorem 4.6) suggests that in such high-complexity scenarios, our topological regularization method should yield more substantial performance gains over the standard baseline. The experimental results clearly corroborate this. We will include the complete experimental setup and training details in the Appendix of the final manuscript.
>
> **Response to W4:**
>
>  Regarding the selection strategy for the parameter $\eta$, please refer to Appendix D.4 and the experiments provided below (branch=128):
>
> | $\eta$  |   0   | 0.01  |    0.1    |    0.5    |   1   |  1.5  |   2   |
> | :-----: | :---: | :---: | :-------: | :-------: | :---: | :---: | :---: |
> | k-NN(%) | 57.70 | 57.95 | **59.10** | **58.83** | 50.35 | 41.55 | 31.44 |
> | Acc(%)  | 58.12 | 58.33 | **60.94** | **60.95** | 57.62 | 50.86 | 47.32 |
>
> The optimal $\eta$ falls between 0.1 and 0.5, which perfectly aligns with our theoretical requirement to maintain connectivity within the $[\alpha, \beta]$ interval. For detailed experiments regarding $\eta$ values between 0.1 and 0.5, please refer to Appendix D.4.
>
> Since intrinsic geometric density naturally varies across classes, making $\eta$ adaptive is a rigorous next step. We are exploring making $\eta$ a function of local topological persistence. For example, an adaptive $\eta_k$ for a local component $C_k$ could dynamically align with the moving average of its dominant death times during training. We will detail these specific ideas in our future work.
>
> [1] Chuang, C. Y., et al. Debiased contrastive learning. NeurIPS, 33: 8765-8775, 2020.
>
> [2] Zhao, X., et al. ArCL: Enhancing Contrastive Learning with Augmentation-Robust Representations. ICLR, 2023.

---

> > ### Author Rebuttal · Reviewer_axD4 · 2026-04-03
> >
> > Thanks for your detailed response. I think most of my concerns are solved and have raised my score to 4. I hope the discussions on differences from related works and the large-scale experiments can be added to the revised paper.

---

> > > ### Author Response · Authors · 2026-04-03
> > >
> > > Thank you very much for raising your score.
> > >
> > > We are glad that our rebuttal has addressed your concerns. We will incorporate a discussion of the differences from related work as well as large-scale experimental results into the revised paper.

---

### Official Review · Reviewer_uHx6 · 2026-03-11

**Soundness:** 3
**Presentation:** 3
**Significance:** 3
**Originality:** 3
**Overall Recommendation:** 4
**Confidence:** 3

**Summary:**

This paper proposes a Topology-Aware Contrastive Learning method. The method leverages persistent homology to model the topological connectivity of the representation space and explicitly regulates the connectivity relationships among samples during training. In this way, positive samples remain compact within classes while preserving a reasonable topological structure, thereby improving the discriminability of the learned representations. From a theoretical perspective, the paper also provides a related generalization analysis. Experimental results show that the proposed method can learn clearer and more structurally consistent feature representations.

**Compliance With Llm Reviewing Policy:**

Affirmed.

**Final Justification:**

Given that the author's response has addressed most of my concerns, I maintain my score.

**Key Questions For Authors:**

1. The topological structures of datasets may vary significantly. Does the proposed method remain effective when dealing with datasets that have more complex structures or higher levels of noise?
2. The compared methods in the paper mainly focus on the SimCLR baseline. Could the authors include more recent or representative contrastive learning methods for comparison to provide a more comprehensive evaluation of the proposed method?
3. When designing the topological constraint, is there a risk of over-constraining the structure of the representation space, which might limit the model’s ability to learn more flexible feature distributions?
4. The citation format for papers is inconsistent. It is recommended to carefully check and unify the reference format.

**Limitations:**

yes

**Strengths And Weaknesses:**

### Strengths
1. It analyzes the limitations of traditional contrastive learning from a topological perspective, pointing out that relying solely on geometric distance optimization may lead to an unreasonable structure in the representation space.
2. The paper introduces persistent homology to model the connectivity structure of the representation space, enabling the model to maintain overall structural information while preserving intra-class compactness.
3. The method is not only designed at the algorithmic level but is also supported by theoretical analysis. It explains how topology-agnostic optimization may lead to representation confusion, thereby strengthening the theoretical foundation of the proposed method.
### Weaknesses
1. Introducing topological computations such as persistent homology may incur additional computational overhead, and the paper provides insufficient analysis regarding the space complexity and scalability of the proposed method.
2. The method involves several hyperparameters related to the topological structure, yet their sensitivity and parameter selection strategy are not systematically discussed.
3. Although the experiments verify the effectiveness of the method, more comprehensive experimental analyses, such as ablation studies, are lacking.

---

> ### Author Rebuttal · Authors · 2026-03-31
>
> Dear Reviewer uHx6,
>
> We are very grateful for your valuable comments. Our responses are as follows:
>
> **Response to W1:**
>
> Our additional computational overhead primarily stems from computing 0-dimensional persistent homology. By leveraging well-established persistent homology libraries, this extra cost remains marginal compared to the baseline.
>
> **Response to W2 and W3:**
>
> Due to the page limit of the main text, we placed the systematic discussion and parameter selection strategies in **Appendix D**. We kindly direct the reviewer's attention to the following sections:
>
> - **Regularization Coefficient ($\lambda$):** In Section D.1, we demonstrate the sensitivity of $\lambda$ across CIFAR-10 and CIFAR-100, showing that performance peaks at $\lambda=0.01$ and degrades at overly large values ($\lambda=0.1$) because excessive regularization overshadows the primary contrastive task.
>
> - **Positive Sample Size ($n$):** In Section D.2, we detail our parameter selection strategy for $n$, justifying $n=4$ as the optimal balance between providing sufficient topological density to approximate local manifold structures and avoiding excessive computational overhead.
>
> - **Branch Dimension ($B$) and Radius ($\eta$):** In Section D.4, we provide a joint sensitivity analysis (Figure 3) through a grid search over $\eta \in \{0.1, 0.2, 0.3, 0.4, 0.5\}$ and $B \in \{8, 16, 32, 64, 128\}$. This analysis shows an inverted U-shaped performance curve, guiding our selection of moderate granularities for optimal semantic retention.
>
> For additional experiments regarding the selection strategy of $\eta$, please refer to our Response to W4 for Reviewer axD4.
>
> **Response to Q1:**
>
> To extend our method to datasets with richer structures, we conducted experiments on the ImageNet-100 and STL-10 datasets. The linear evaluation accuracy and k-NN Purity (k=3) are shown in the table below:
>
> |   Dataset    | k-NN (SimCLR) | k-NN (Ours) | Acc (SimCLR) | Acc (Ours) |
> | :----------: | :-----------: | :---------: | :----------: | :--------: |
> | ImageNet-100 |     30.59     |  **37.19**  |    60.98     | **64.14**  |
> |    STL-10    |     74.59     |  **75.27**  |    82.47     | **83.21**  |
>
> As can be seen, our method remains effective when handling data with more complex structures, especially for datasets like ImageNet-100, where the inherent risks of semantic overlap and manifold entanglement are already high.
>
> **Response to Q2:**
>
> To comprehensively demonstrate the superiority of our approach, we conducted additional experiments on CIFAR-100 to compare standard SimCLR, ArCL[1]，DCL[2], and our proposed method. We evaluated their impact on the latent space structure (measured via k-NN purity) and their sensitivity to extreme augmentation strategies (Strong vs. Weak augmentations, as defined in Section 6.2; here, we only present the accuracy under weak and strong photometric augmentations).
>
> |  Method  | k-NN Purity (%) |  S-Top1   |  S-Top5   |  W-Top1   |  W-Top5   |
> | :------: | :-------------: | :-------: | :-------: | :-------: | :-------: |
> |  SimCLR  |      57.73      |   11.33   |   31.45   |   9.10    |   25.03   |
> |   DCL    |      57.77      |   12.61   |   34.22   |   6.37    |   20.06   |
> |   ArCL   |      59.77      |   12.51   |   32.90   |   10.08   |   27.04   |
> | **Ours** |    **60.36**    | **17.35** | **39.49** | **12.56** | **32.18** |
>
> It can be observed that our method is significantly more robust under extreme data augmentation strategies compared to the other baseline methods.
>
> **Response to Q3:**
>
> it is actually standard contrastive learning that risks over-constraining the space. Standard methods implicitly force positive pairs to a geometric distance of zero ($d(z, z^+) \to 0$). This strict singularity aggressively eliminates intra-class variance and leads to the "topology-agnostic confusion".
>
> In contrast, our Topology-Aware objective explicitly _relaxes_ this strict geometric constraint. By enforcing an $\alpha-\beta$ connectivity , we require positive samples to simply remain within the same connected topological component rather than overlapping at a single coordinate. As detailed in Section 3.3, our bi-directional indicator function penalizes distances that are too large (to ensure connectivity) and too small (to maintain separability). Therefore, our method actively preserves flexible feature distributions.
>
> **Response to Q4:**
>
> We appreciate the reviewer pointing this out. We will carefully proofread the bibliography and strictly unify all reference formats in the camera-ready version.
>
> [1] Chuang, C. Y., et al. Debiased contrastive learning. NeurIPS, 33: 8765-8775, 2020.
>
> [2] Zhao, X., et al. ArCL: Enhancing Contrastive Learning with Augmentation-Robust Representations. ICLR, 2023.

---

> > ### Author Rebuttal · Reviewer_uHx6 · 2026-04-03
> >
> > I thank the authors for their response, which has partially addressed my concerns. My follow-up questions are as follows:
> >
> > Regarding W1, the analysis of the proposed method's space complexity and scalability remains insufficient and unconvincing.
> >
> > For W3, the more comprehensive experimental analysis mentioned (such as ablation studies) still needs to be provided.
> >
> > In terms of baseline comparisons, it seems necessary to include more state-of-the-art methods (the current baselines from 2020 to 2023 are not recent enough).
> >
> > The author addressed my concerns mentioned above in a new response. At AC's request, I have changed the rebuttal acknowledgement from (b) to (a).

---

> > > ### Author Response · Authors · 2026-04-04
> > >
> > > Dear Reviewer axD4,
> > >
> > > Thank you for your constructive follow-up comments. We have addressed your remaining concerns below:
> > >
> > > **Space Complexity Analysis:**
> > >
> > > In standard contrastive learning, computing the InfoNCE loss requires calculating cosine similarities between all sample pairs within a batch of size $N$. Since each sample generates $n$ views, the model processes $n \cdot N$ samples and must construct an $(n \cdot N) \times (n \cdot N)$ similarity matrix, yielding a space complexity of $O((n \cdot N)^2)$.
> > >
> > > In our proposed topological regularization module, however, calculating the $H_0$ persistent homology for the $n$ points only requires computing a local distance matrix for the $n$ views of each anchor. This merely requires allocating an $n \times n$ distance matrix per anchor, resulting in a minor space overhead of $O(N \cdot n^2)$. When combined with the standard contrastive objective, the overall asymptotic behavior is dominated by the highest-order term. Consequently, the overall space complexity of our TACL framework remains $O((n \cdot N)^2)$.
> > >
> > >
> > > **Scalability Analysis:**
> > >
> > > Regarding data scalability, we demonstrated our model's sustained advantage on high-resolution datasets including CIFAR-10, CIFAR-100, ImageNet-100, and STL-10 (detailed in the subsequent ablation study). For odel scalability, we validated our approach on both ResNet-18 and ResNet-50, proving its effectiveness is agnostic to model scale. Finally, regarding computational scalability, the table below shows that our topological regularization introduces only a marginal memory overhead and extra training time per epoch(evaluated on the CIFAR-100). This confirms the high practical scalability of TACL.
> > >
> > > |    Method     | Backbone  | Memory (GB)  | Time / Epoch (s) |
> > > | :-----------: | :-------: | :----------: | :--------------: |
> > > |    SimCLR     | ResNet-18 |      7       |        12        |
> > > | SimCLR + TACL | ResNet-18 | **8(+11%)**  |  **18 (+15%)**   |
> > > |    SimCLR     | ResNet-50 |      28      |        27        |
> > > | SimCLR + TACL | ResNet-50 | **32(+11%)** |  **36 (+13%)**   |
> > >
> > >
> > > **Ablation Study:**
> > >
> > > Our ablation study analyzes the sensitivity of $\lambda$, where $\lambda=0$ represents the baseline without topological regularization, and $\lambda=0.01$ yields optimal performance (see Appendix D.1). We also evaluate optimizing the model solely with the topological term. Furthermore, we expand our ablation to ImageNet-100 and STL-10 to assess both overall performance and topological purity (via $k$-NN accuracy).
> > >
> > > |           Acc(%)            | CIFAR-10  | CIFAR-100 | ImageNet-100 |  STL-10   |
> > > | :-------------------------: | :-------: | :-------: | :----------: | :-------: |
> > > |           SimCLR            |   84.33   |   60.38   |    60.98     |   82.47   |
> > > |            TACL             |   22.56   |   18.98   |    10.65     |   15.07   |
> > > | SimCLR+TACL（$\lambda=0.01$） | **85.47** | **62.47** |  **64.14**   | **83.21** |
> > >
> > > |       k-NN Purity(%)        | CIFAR-10  | CIFAR-100 | ImageNet-100 |  STL-10   |
> > > | :-------------------------: | :-------: | :-------: | :----------: | :-------: |
> > > |           SimCLR            |   71.24   |   57.73   |    30.59     |   74.59   |
> > > |            TACL             |   10.56   |   9.57    |     9.53     |   11.56   |
> > > | SimCLR+TACL（$\lambda=0.01$） | **71.77** | **60.36** |  **37.19**   | **75.27** |
> > >
> > > **Comparison with Recent Methods**
> > >
> > > We compare our method with recent baselines, including NCL [1], which constrains the latent space to the non-negative orthant; Kernel-InfoNCE [2], which implicitly performs spectral clustering via kernel functions; and $\mathbb{X}$-CLR [3], which introduces soft-graph repulsive forces based on relative sample relationships. Notably, none explicitly incorporate topological perspectives into contrastive learning, nor account for robustness under extreme augmentations. Therefore, we evaluate performance from two primary perspectives: 1) latent space structure (measured by $k$-NN purity), and 2) robustness under weak and strong photometric augmentations (detailed in Section 6.2).
> > >
> > > |      Method      | k-NN Purity (%) |  S-Top1   |  S-Top5   |  W-Top1   |  W-Top5   |
> > > | :--------------: | :-------------: | :-------: | :-------: | :-------: | :-------: |
> > > |      SimCLR      |      57.73      |   11.33   |   31.45   |   9.10    |   25.03   |
> > > |       NCL        |      57.23      |   13.89   |   34.74   |   8.16    |   23.30   |
> > > |  Kernel-InfoNCE  |      54.10      |   2.43    |   9.99    |   2.33    |   8.65    |
> > > | $\mathbb{X}$-CLR |      56.27      |   11.98   |   31.44   |   6.92    |   22.43   |
> > > |     **Ours**     |    **60.36**    | **17.35** | **39.49** | **12.56** | **32.18** |
> > >
> > > [1] Wang, Y., et al. Non-negative contrastive learning. ICLR, 2024.
> > >
> > > [2] Tan, Z., et al. Contrastive Learning is Spectral Clustering on Similarity Graph. ICLR, 2024.
> > >
> > > [3] Sobal, V., et al. $\mathbb{X}$-Sample Contrastive Loss: Improving Contrastive Learning with Sample Similarity Graphs. ICLR, 2025.

---

### Official Review · Reviewer_J2ga · 2026-03-11

**Soundness:** 2
**Presentation:** 2
**Significance:** 2
**Originality:** 2
**Overall Recommendation:** 3
**Confidence:** 4

**Summary:**

In standard contrastive learning, the distance between positive pairs is minimized. However, because geometric information in the latent feature space is not taken into account, clusters that should be semantically separated may end up having large overlapping regions. To address this issue, the paper introduces persistent homology and analyzes the connectivity of clusters, proposing a contrastive learning method that reduces such overlapping regions.

**Compliance With Llm Reviewing Policy:**

Affirmed.

**Key Questions For Authors:**

Please address the weaknesses listed.

**Limitations:**

Yes

**Strengths And Weaknesses:**

**Strength:**
Since geometric information in the latent feature space is not considered, clusters that should be semantically distinct may end up sharing large overlapping regions. The authors analyze the latent feature space from the perspective of connectivity and propose a learning method that reduces the shared regions between semantically different clusters. They propose a new contrastive learning method based on a geometric perspective and provide several theoretical guarantees.

**Weakness:**
- How frequently does the situation shown in Fig. 1(a) occur? The experimental section provides an example involving cars and motorcycles, but does the same phenomenon occur in other datasets as well?

- How are the parameters $\alpha$ and $\beta$ determined?

- I do not understand how to interpret Fig. 2. Are these differences statistically significant? What do the orange and blue shapes behind the plots represent?

- Is there a clear connection between the proven theoretical results and the experimental results? It does not appear that the theoretical results lead to a significant improvement in the experiments.

- How does the computational time compare to the baseline? How much additional computation time is required to achieve the reported improvements?

- Several theorems are proved regarding the theoretical guarantees of the proposed method, but they seem to be combinations of well-known results in statistical machine learning, making it difficult to see true novelty.

- Theorem 5.2: What are the definitions of $L$ and $\kappa$?

- Table 1: It does not appear that using persistent homology leads to a large improvement in accuracy.


- (Minor comment) Theorem 4.6: “$\exist$” is unnecessary.

 -Trofimov et al., 2023 and Moor et al., 2020 do not seem appropriate as background references for persistent homology. More suitable references should be cited.

---

> ### Author Rebuttal · Authors · 2026-03-31
>
> Dear Reviewer J2ga,
>
> We are very grateful for your valuable comments. Our responses are as follows:
>
> **Response to W1:**
>
> The frequency of Topology-Agnostic Confusion (TAC) heavily depends on **dataset characteristics and augmentation strategies**. As shown in App. D.3, coarse-grained datasets (e.g., CIFAR-10) possess larger inherent geometric margins, maintaining a higher topological purity (71.24% in the baseline) with a lower risk of semantic overlap. Conversely, fine-grained datasets (e.g., CIFAR-100) are highly susceptible to TAC due to visually similar subclasses. Regarding data augmentations, varying strengths inevitably induce feature collapse and semantic confusion. As proven in Theorem 4.6, these variations significantly increase the risk of TAC.
>
> **Response to W2:**
>
> We clarify that $\alpha$ and $\beta$ are not manually set hyperparameters, but boundary values that naturally emerge during training dynamics. The only explicitly set hyperparameter is the target connectivity radius $\eta$ (App. D.4). Our optimization combines semantic alignment (InfoNCE) with topological regularization (connectivity loss). Because InfoNCE tends to pull positive pairs toward a single point (geometric collapse) while the topology loss strives to maintain distance $\eta$, positive pairs do not converge perfectly. Instead, they fall into an interval $[\alpha, \beta]$ centered around $\eta$, which underpins our theoretical analysis regarding collapse and confusion.
>
> **Response to W3:**
>
> The TAC phenomenon illustrated in Figure 2 is significant (evidenced by the low $k$-NN purity of the five randomly selected superclasses in Table 1). We simply selected the "Vehicles 1" superclass for illustrative purposes. The orange and blue shaded areas plotted along the top and right axes represent the marginal probability density distributions.
>
> **Response to W4:**
>
> We outline how our theoretical framework drives the empirical design: Theorem 4.6 proves the high probability of TAC occurring based on our framework, a phenomenon also observed empirically (Fig. 2). Theorem 4.7 derives a generalization bound incorporating the effective number of connected neighbors $C$, providing the theoretical foundation that richer connectivity effectively tightens downstream risk. Our experiments (Table 1) confirm that our method mitigates TAC, yielding purer semantic clusters. Theorem 5.2 proves our method effectively mitigates feature collapse ($R_\epsilon$) and semantic drift ($R_\delta$), which is directly verified by Table 2.
>
> **Response to W5:**
>
> Our additional computational overhead primarily stems from computing 0-dimensional persistent homology. By leveraging well-established persistent homology libraries, this extra cost remains marginal compared to the baseline.
>
> **Response to W6:**
>
> Our **core novelty** lies in constructing a persistent homology-based topological framework to decode contrastive learning, distinguishing our work from augmentation overlap[1] and spectral[2] perspectives. Based on this, we formalize TAC (Theorem 4.6) and propose a solution grounded in a connectivity-dependent generalization bound (Theorem 4.7)—explicitly incorporating a topological loss ($\mathcal{L}_\eta$) to increase the effective number of connected neighbors ($C$). Furthermore, we establish a unified measure-theoretic augmentation sensitivity analysis (Theorem 5.2), proving that our method renders the model highly robust to semantic drift and suboptimal augmentations.
>
> **Response to W7:**
>
> $L$ is the Lipschitz constant of the continuous data augmentation function $A_{\theta}(x)$ w.r.t. its continuous parameter $\theta$. As defined prior to Theorem 5.2, $\|A_{\theta_1}(x) - A_{\theta_2}(x)\| \le L\|\theta_1 - \theta_2\|$. **$\kappa$** is the Lipschitz constant of the feature encoder $f$, satisfying $\|f(x_1) - f(x_2)\| \le \kappa \|x_1 - x_2\|$.
>
> **Response to W8:**
>
> Table 1 validates our theoretical claim that persistent homology prevents TAC and preserves structural integrity. For "Vehicles 1" at $k=3$, purity improves from 64.90% to 67.30%, demonstrating our method successfully pulls semantically similar instances closer while maintaining inter-class separation. The most significant accuracy improvement appears in Table 2, which tests sensitivity to augmentation strength. For instance, under "strong photometric" settings, our topology-aware approach yields a 6% absolute performance boost. Overall, our method makes the model highly robust to semantic drift and suboptimal data augmentations.
>
> **Response to W9:**
>
> We will remove the $\exists$ symbol for mathematical rigor. We will also reposition the references for Trofimov et al. (2023) and Moor et al. (2020).
>
> [1] Wang, Y., et al. An Augmentation Overlap Theory of Contrastive Learning. JMLR, 26(228): 1-42, 2025.
>
> [2] HaoChen, J. Z., et al. Provable guarantees for self-supervised deep learning with spectral contrastive loss. NeurIPS, 34:5000–5011, 2021.

---

> > ### Author Rebuttal · Reviewer_J2ga · 2026-04-04
> >
> > I thank the authors for their rebuttal. However, my concerns remain unresolved. The use of 0-dimensional homology appears closely related to standard clustering structure, and the rebuttal does not convincingly demonstrate sufficient novelty or practical benefit relative to its computational overhead. The empirical gains are also modest. I will therefore keep my score unchanged.

---

> > > ### Author Response · Authors · 2026-04-04
> > >
> > > We sincerely thank the reviewer for the reply. Here, we further address your remaining concerns:
> > >
> > > First, we clarify that 0-dimensional persistent homology ($H_0$) provides a quantitative, differentiable proxy for connectivity evolution, not a static partition like standard clustering [1]. Classical clustering operates on fixed representations to produce discrete assignments. In contrast, $H_0$ tracks how connected components merge across continuous scales, effectively capturing the stability of local neighborhoods under representation perturbations. This is crucial for contrastive learning, because false positives fundamentally stem from unstable connectivity across different data augmentations.
> > >
> > > We apologize that, due to space limitations in the first rebuttal round, we were unable to provide the detailed computational overhead table. As demonstrated below, our topological regularization introduces only marginal memory overhead and minimal extra training time per epoch (evaluated on the CIFAR-100).
> > >
> > > |**Method**|**Backbone**|**Memory (GB)**|**Time / Epoch (s)**|
> > > |---|---|---|---|
> > > |SimCLR|ResNet-18|7|12|
> > > |**SimCLR + TACL**|**ResNet-18**|**8 (+11%)**|**18 (+15%)**|
> > > |SimCLR|ResNet-50|28|27|
> > > |**SimCLR + TACL**|**ResNet-50**|**32 (+11%)**|**36 (+13%)**|
> > >
> > > Finally, regarding the empirical gains, the frequency of Topology-Agnostic Confusion heavily depends on dataset characteristics. In coarse-grained datasets (e.g., CIFAR-10), which inherently possess larger intrinsic geometric margins, the baseline already maintains high topological purity. Consequently, the actual benefits appear relatively modest compared to fine-grained datasets. However, on fine-grained datasets (e.g., CIFAR-100, ImageNet-100) where Topology-Agnostic Confusion occurs with a much higher probability, the performance improvements delivered by our method are highly considerable.
> > >
> > > [1] Edelsbrunner H, Harer J. Computational topology: an introduction. American Mathematical Soc., 2010.

---

### Official Review · Reviewer_Vqn7 · 2026-03-11

**Soundness:** 3
**Presentation:** 2
**Significance:** 3
**Originality:** 2
**Overall Recommendation:** 5
**Confidence:** 3

**Summary:**

The paper introduces a Topology-Aware Contrasting Learning framework that leverages Persistent Homology information from the Vietoris-Rips filtration constructed from the data's latent representations to explicitly regularize the connectivity of the latent representations and balance intra-class cohesion with separability. The motivation for the method comes from the highlighted topology-agnostic confusion phenomenon, defined as the inter-class confusion between latent representations of different classes that can happen due to manifold entanglement and latent collapse as a result of ignoring the intrinsic structure of the data and the topological complexity, especially when semantic drift happens due to strong data augmentations. The authors formally define the phenomenon with a lower bound on the probability of class confusion based on the degree of feature collapse, and introduce α–β connectivity to explain how the alignment of positive pairs shapes the latent space. Using these, they explain how topological constraints relate to class confusion and derive how topological connectivity quantifies the generalization error in both unsupervised pre-training and supervised downstream training. They further introduce a measure-theoretic framework for analyzing augmentation sensitivity and derive a bound on augmentation-induced errors based on α–β connectivity.

The authors identify the limitations of standard contrastive learning through theoretical analysis, construct a Vietoris-Rips filtration on the latent representations, and extract the 0-dimensional persistent homology death times as representatives of the critical thresholds defined by the lengths of edges in the latent complex that trigger the merge of distinct components. These values are then used to obtain a topology-aware contrastive loss that controls the connectivity of latent representations, penalizing large distances to maintain connectivity while penalizing very small distances to preserve separability, used together with the InfoNCE loss. Experiments are performed on CIFAR-100 using a ResNet-50 backbone. The method is compared to SimCLR, evaluated using the accuracy of a fine-tuned linear classifier with a frozen pre-trained encoder and k-NN purity scores (the proportion of neighbors sharing the same class label among their 3, 8, and 20 nearest neighbors) on the full CIFAR-100 and five randomly selected CIFAR-100 subclasses. In addition, the method is compared with SimCLR across different weak and strong augmentation scenarios using linear classifier accuracy, and further experiments are conducted to demonstrate its performance on CIFAR-10. In general, the results indicate better separation of class clusters and improvements in accuracy and k-NN purity in all cases, with larger improvements on CIFAR-100 than on CIFAR-10. These support the paper's claims and theoretical results.

**Compliance With Llm Reviewing Policy:**

Affirmed.

**Final Justification:**

My main concerns were the originality of the work and the limited evaluation setting. The extended discussion of how their work differs from existing work in topological contrastive learning, along with the additional experiments presented during the rebuttal phase, addresses my main concerns, and I raised my score to 5.

**Key Questions For Authors:**

- My main question is whether it would be possible to discuss the existing literature on topological deep learning and contrastive learning, and to incorporate this into the paper while clearly distinguishing how it differs from existing approaches. This would strengthen the claims and significantly improve the paper.
For this question, the following papers: "Improving Self-supervised Molecular Representation Learning using Persistent Homology" by Luo et al. in NeurIPS 2023, and "TopoGCL: Topological Graph Contrastive Learning" by Chen et al. in AAAI 2024 are of interest, and the papers "Hyperbolic Contrastive Learning for Visual Representations beyond Objects" by Ge et al. in CVPR 2023, and "Learning Topology-Preserving Data Representations" by Trofimov et al. in ICLR 2023 are relevant as well. Note that the paper by Trofimov is cited/mentioned, but it can be more clearly distinguished.
- Have you tested the method on any other dataset or data modality besides 2D images, or on any model other than ResNet50, to ensure the experimental results are generalizable to other settings or models and are not limited to this scenario?

**Limitations:**

Yes

**Strengths And Weaknesses:**

Soundness:
- Claims are well supported by the theoretical results, and the proofs are based on reasonable assumptions. The intuitions for the theorems are well explained.
- The experimental setting is well-designed to analyze the approach; however, it is limited in terms of data, model, and baseline variation (only CIFAR-10/CIFAR-100, only ResNet50, only SimCLR). More variation in one or more of these would significantly strengthen the claims.

Presentation:
- The submission is well-written, and the narrative is easy to follow. The results are well-presented and discussed.
- Although previous contrastive learning algorithms and the usage of persistent homology in machine learning are discussed, the paper would benefit significantly from a discussion of other Topological Contrastive Learning methods and how this method differs from them.

Significance:
- The paper could influence future research on contrastive learning and other areas that learn representations of data. The scope of impact is broad and appropriate for contribution. However, the current evaluation focuses on a narrow part of this impact, mostly on one specific dataset-model combination, and compared to one model, extending this to more varied settings would strengthen the impact.
- It advances understanding of machine learning and its capabilities by bringing out the geometric limitations of standard contrastive learning approaches.

Originality:
- The paper provides new insights into the key properties and challenges of existing contrastive learning approaches. It provides a novel approach, supported by theoretical and experimental analysis.
- Contributions are well defined, but a discussion of existing topological contrastive learning approaches and how they differ is needed to distinguish them better from closely related literature.

---

> ### Author Rebuttal · Authors · 2026-03-31
>
> Dear Reviewer Vqn7,
>
> We are very grateful for your valuable comments. Our responses are as follows:
>
> **Response to Q1:**
>
> We thank the reviewer for these excellent references. We will discuss them in the revision to highlight our unique contributions. Specifically, our work differs as follows:
>
> - **Compared to Luo et al. [1] :**
>
> 	While this work also introduces topological concepts, it focuses on Graph Contrastive Learning by extracting topological features directly from structured inputs. Our framework operates differently: rather than relying on structural priors, we dynamically apply Persistent Homology to the Euclidean latent space. Specifically, we compute the Vietoris-Rips filtration on mini-batches of augmented views to regulate how the model constructs its representations. This allows our method to handle unstructured data (like images), where input-level topology is heavily obscured by pixel-level noise.
>
> - **Compared to Chen et al. [2] :**
>
>     Similarly, TopoGCL focuses on structured graph data. It leverages Persistent Homology to directly extract topological feature representations for contrastive learning. Conversely, we use Persistent Homology solely as a mechanism to determine which sample pairs are structurally critical (i.e., critical edges) within the latent space.
>
> - **Compared to Ge et al. [3] :**
>
>     This work tackles the limitations of standard Euclidean contrastive learning by altering the underlying geometry—mapping features to hyperbolic space to better capture hierarchical, tree-like relationships. We do not change the base geometry from Euclidean to hyperbolic; rather, we impose an explicit topological constraint within standard Euclidean space. While Ge et al. aim to embed a hierarchical taxonomy, our goal is to regulate 0-dimensional homology ($H_0$) to manage connected components. We specifically address the "topology-agnostic confusion" caused by augmentation overlap, ensuring that distinct manifolds do not entangle when strong augmentations are applied.
>
> - **Compared to Trofimov et al. [4] :**
>
> 	Trofimov et al. focus on topology-preserving representations by matching latent persistence diagrams with the input manifold. We do not preserve input topology, as pixel-space topology lacks semantic meaning for classification. Instead, we actively sculpt a new latent topology. By regulating the "death times" of augmented views to a target radius $\eta$, we enforce a specific $\alpha-\beta$ connectivity. This addresses a key InfoNCE vulnerability: preventing zero-distance feature collapse that allows semantic drift to build spurious bridges between distinct classes.
>
> **Response to Q2:**
>
> Your suggestion regarding extending our method to other modalities is excellent. Frankly speaking, as our initial scientific insight—the observation of "topology-agnostic confusion" —was rooted in analyzing image data behavior under strong augmentations, the empirical portion of our current work is focused on the 2D image domain. However, our theoretical framework is inherently modality-agnostic. The Vietoris-Rips filtration solely requires computing pairwise distances $d(z_i, z_j)$ within a mini-batch. This implies that any modality relying on data augmentation to generate positive pairs (e.g., time-series or text data) can directly incorporate our connectivity regularizer. We will explicitly emphasize this theoretical cross-modality potential in the "Discussion" section to guide future research.
>
> Regarding the architectural generalization you mentioned, we have conducted additional pre-training experiments using a ResNet-18 backbone. We will include a new table in the Appendix to demonstrate that our method is architecture-independent; whether the encoder employs local convolutions or global attention mechanisms, enforcing $\alpha-\beta$ connectivity consistently yields gains in both topological purity and linear evaluation accuracy.
>
> | Architecture | k-NN (SimCLR) | k-NN (Ours) | Acc (SimCLR) | Acc (Ours) |
> | :----------: | :-----------: | :---------: | :----------: | :--------: |
> |   Resnet18   |     38.37     |  **41.07**  |    52.02     | **55.06**  |
>
> Furthermore, to demonstrate the dataset generalization of our approach, we have also conducted experiments on ImageNet-100 and STL-10. For the specific experimental results, please refer to our response to Reviewer uHx6 (Q1).
>
> [1] Luo Y., et al. Improving Self-supervised Molecular Representation Learning using Persistent Homology. NeurIPS, 36: 34043-34073, 2023.
>
> [2] Chen, Y., et al. TopoGCL: Topological Graph Contrastive Learning. AAAI, 38(10): 11453-11461, 2024.
>
> [3] Ge, S., et al. Hyperbolic Contrastive Learning for Visual Representations beyond Objects. CVPR, 6840-6849, 2023.
>
> [4] Trofimov, I., et al. Learning Topology-Preserving Data Representations. ICLR, 2023.

---

> > ### Author Rebuttal · Reviewer_Vqn7 · 2026-04-03
> >
> > I thank the authors for their detailed response. With additional experiments on ResNet18, ImageNet, and STL, and a more in-depth discussion of related work, my concerns have been adequately addressed, and I am updating my score to 5.

---

> > > ### Author Response · Authors · 2026-04-03
> > >
> > > Thank you very much for raising your score. We are glad that our rebuttal has addressed your concerns.

---

### Decision · Program_Chairs · 2026-04-30

**Decision:**

Accept (regular)

**Comment:**

This paper proposes a mechanism for making contrastive learning application "topology-aware." This is achieved via regularizing connectivity in the latent space by using persistent homology, a technique from computational topology. The paper is technically sound and relevant; all reviewers agreed on this fact. During the rebuttal phase, concerns about the scalability and the "depth" of the experiments were raised; a comprehensive rebuttal helped alleviate these concerns.

While there is still some work left to do, specifically the integration of the new experiments and discussions promised by the authors during the rebuttal phase, I am happy to recommend this submission for presentation at the conference. My somewhat weaker overall endorsement is based on the fact that the method is in fact a direct application of the method by [Hofer et al.](https://proceedings.mlr.press/v97/hofer19a.html); this is partially mitigated by the theoretical analysis provided in this submission, though, but, overall, scalability still remains a concern.